evolution, genetics

arctic, climate change, genetic diversity, ice-dependent, sea ice loss, *Ursus maritimus*

**Authors for correspondence:**
Simo Njabulo Maduna
e-mail: simo.maduna@nibio.no
Snorre B. Hagen
e-mail: snorre.hagen@nibio.no

# Sea ice reduction drives genetic differentiation among Barents Sea polar bears

Simo Njabulo Maduna[1], Jon Aars[2], Ida Fløystad[1], Cornelya F. C. Klütsch[1], Eve M. L. Zeyl Fiskebeck[3], Øystein Wiig[3], Dorothee Ehrich[4], Magnus Andersen[2], Lutz Bachmann[3], Andrew E. Derocher[5], Tommi Nyman[1], Hans Geir Eiken[1] and Snorre B. Hagen[1]

[1]Norwegian Institute of Bioeconomy Research, Division of Environment and Natural Resources, Svanhovd, N-9925 Svanvik, Norway
[2]Norwegian Polar Institute, N-9296 Tromsø, Norway
[3]Natural History Museum, University of Oslo, N-0318 Oslo, Norway
[4]Department of Arctic and Marine Biology, UiT Arctic University of Tromsø, N-9037 Tromsø, Norway
[5]Department of Biological Sciences, University of Alberta, Edmonton, Alberta, Canada T6G 2E9

SNM, 0000-0002-9372-4360; JA, 0000-0002-0574-6712; IF, 0000-0002-0484-4265;
CFCK, 0000-0001-8238-2484; EMLZF, 0000-0002-6858-1978; ØW, 0000-0003-0395-5251;
DE, 0000-0002-3028-9488; MA, 0000-0001-7024-5466; LB, 0000-0001-7451-2074;
AED, 0000-0002-1104-7774; TN, 0000-0003-2061-0570; HGE, 0000-0002-5368-3648;
SBH, 0000-0001-8289-7752

Loss of Arctic sea ice owing to climate change is predicted to reduce both genetic diversity and gene flow in ice-dependent species, with potentially negative consequences for their long-term viability. Here, we tested for the population-genetic impacts of reduced sea ice cover on the polar bear (*Ursus maritimus*) sampled across two decades (1995–2016) from the Svalbard Archipelago, Norway, an area that is affected by rapid sea ice loss in the Arctic Barents Sea. We analysed genetic variation at 22 microsatellite loci for 626 polar bears from four sampling areas within the archipelago. Our results revealed a 3–10% loss of genetic diversity across the study period, accompanied by a near 200% increase in genetic differentiation across regions. These effects may best be explained by a decrease in gene flow caused by habitat fragmentation owing to the loss of sea ice coverage, resulting in increased inbreeding of local polar bears within the focal sampling areas in the Svalbard Archipelago. This study illustrates the importance of genetic monitoring for developing adaptive management strategies for polar bears and other ice-dependent species.

## 1. Introduction

Climate change is rapidly altering the structure, dynamics and functioning of ecosystems, leading to large-scale changes in the distribution, demography and phenology of species [1]. The current warming trend is fastest in the Arctic, causing a reduction of the extent, thickness, multiyear persistence and seasonal duration of sea ice cover [2]. Species relying on ice-habitats for foraging, reproduction and movement are therefore particularly vulnerable [3–6]. Population declines as well as contraction and fragmentation of geographical ranges have indeed been documented in ice-dependent species such as the Adélie penguin (*Pygoscelis adeliae*) and chinstrap penguin (*Pygoscelis antarcticus*, [7,8]), Baltic ringed seal (*Pusa hispida botnica*, [9]) and polar bears (*Ursus maritimus*, [10–13]).

Detrimental ecological and demographic effects of reductions in sea ice cover may ultimately reduce the standing genetic variation of species [14–17]. Loss of genetic diversity over time (genetic erosion) could introduce an additional level

of adversity for affected species, by diminishing the adaptive potential enabling species to respond to anthropogenic pressures, pathogen outbreaks and environmental change [15,18]. Documenting changes in genetic diversity, spatial population structure and exchange in response to long-term climatic trends is therefore crucial for predicting the future fates of species [15,18]. Nevertheless, direct assessments of climate-induced temporal changes of intraspecific genetic diversity and population connectivity of species affected by reductions in sea ice cover remain scarce. This is largely owing to a lack of coordinated long-term ecological sampling and monitoring efforts [18,19]. In addition, it is unknown if spatial climate gradients, which are often used as proxies to predict responses to environmental change, fully capture the underlying population-genetic processes [18].

Our analysis focused on the polar bears inhabiting the Svalbard Archipelago in the northwest Barents Sea, because there is long-term genetic monitoring data available for this subpopulation going back until 1995. Moreover, the reduction of sea ice cover for the area inhabited by the Barents Sea polar bear subpopulation is an ongoing process. This includes both an estimated increase of ice-free days by 41 per decade between 1979 and 2014 [20] and a documented northward shift in the distribution of optimal habitat for polar bears in all seasons [21]. Coinciding with this period of sea ice loss, studies in Svalbard have revealed both reduced numbers of pregnant females reaching traditional denning areas than before [22,23] and that polar bears spent less time at glacier fronts hunting seals and more time on land and near bird colonies, eating birds and bird eggs, than they did in earlier years [24–26].

Most polar bears in the Barents Sea subpopulation, which was estimated to be approximately 2650 (95% confidence interval (CI) 1900–3600) individuals in August 2004 [27], hunt on pack ice (marginal ice zone) and have been termed 'pelagic' [28]. In summer and autumn, when there is no continuous sea ice cover surrounding Svalbard, roughly 10% of the subpopulation (approx. 250 individuals) occur in the Svalbard Archipelago, and 'local' polar bears that stay in this area year-round predominate [27,29]. In winter and in spring, however, pelagic bears also occur in this area. Thus, depending on the presence of sea ice, there is marked seasonal variation in both the density of bears and the proportion of bears with different movement strategies across the Svalbard Archipelago. Consequently, sea ice reduction is predicted to reduce both the magnitude and duration of the seasonal influx of pelagic bears [30,31]. Importantly, the loss of sea ice habitat in Svalbard in spring, during the mating season of the local polar bears [32], has been more profound than in the remaining Barents Sea areas occupied by the subpopulation, and the loss here has continued in recent years [21]. This increase in ice-free days could reduce mating opportunities and long-range breeding dispersal among regions connected by sea ice and lead to a higher proportion of local mating owing to population fragmentation.

The Svalbard Archipelago has undergone over two decades of rapid sea ice loss [20]. To assess whether and how the declining yearly sea ice season in the Svalbard Archipelago impacts population-genetic parameters of polar bears, we applied an extensive temporal and spatial sampling design to estimate the direction and rate of change of genetic diversity and differentiation in Svalbard polar bears over two decades (1995–2016). Given that the generation time of polar bears is approximately 12 years (International Union for Conservation of Nature/ Species Survival Commission Polar Bear Specialist Group

2015), the study covers at least two generations, in particular if one takes into account that the individuals sampled in the early years of the interval were born up to 20 years earlier (year of birth ranged from 1975 to 2015). Temporally, as both sea ice habitat and connectivity among areas used by pelagic and local bears are in a decline, our hypothesis was that 'local' bears would become increasingly isolated, and that this would be reflected as increasing differentiation among and decreasing genetic diversity within areas. Spatially, because the extent of ice loss is uneven across Svalbard, with the west coast of Spitsbergen (the most western island) showing the highest loss [25], we expected to observe the strongest effects in northwestern Svalbard, where the great majority of bears are 'local' [29].

# 2. Material and methods

## (a) Study area, sample collection and genetic methods

Polar bears from the Barents Sea were captured throughout the Svalbard Archipelago from 1995 to 2016 by the Norwegian Polar Institute, Tromsø, Norway, following standard immobilization, sampling and handling procedures [33] (figure 1). We extracted total genomic DNA from collected tissue samples using the DNeasy Blood & Tissue Kit (Qiagen) following the manufacturer's protocol. For genetic typing, we used 22 published nuclear microsatellite loci with polymerase chain reaction protocols optimized for seven multiplex assays (electronic supplementary material, appendix S1, tables S1–S3). The overall microsatellite data consisted of 626 unique polar bear genotypes, including 206 bears that were previously genotyped following a similar protocol (see the electronic supplementary material, appendix S1). Based on sampling location, we allocated the genotyped bears to four geographical areas: north-western Svalbard (NWS, $n = 123$), northeastern Svalbard (NES, $n = 110$), southwestern Svalbard (SWS, $n = 241$) and southeastern Svalbard (SES, $n = 152$) (see the electronic supplementary material, appendix S1). For each of the four areas, we divided the 22-year sampling period into five temporal groups: T1: 1995–1999; T2: 2000–2004; T3: 2005–2009; T4: 2010–2014; T5: 2015–2016, but excluded periods with fewer than 10 individuals (electronic supplementary material, appendix S1, table S1-1). Thus, our final dataset comprised 16 spatio-temporal groups and 622 individuals, although to explore our results we also evaluated other plausible temporal divisions (electronic supplementary material, appendix S1, table S1-2).

## (b) Spatio-temporal patterns of genetic diversity, inbreeding and differentiation

To facilitate tests of whether population-genetic changes in polar bears coincide with declining sea ice coverage in the Svalbard Archipelago, we estimated several standard parameters of genetic diversity (number of alleles per locus, $A_N$; observed heterozygosity, $H_O$; unbiased expected heterozygosity, $H_E$; allelic richness, $A_R$; and private allelic richness, $A_P$), inbreeding (mean inbreeding coefficient, $F_{IS}$) and relatedness (mean relatedness coefficient, $r$). Moreover, to allow tests of a potential negative association between habitat fragmentation owing to loss of sea ice cover and genetic exchange among Svalbard polar bears, we estimated pairwise genetic differentiation indices (pairwise fixation, $G''_{ST}$; and allelic differentiation, $D_{EST}$) per sampling site and period (for a detailed description of parameters, see the electronic supplementary material, appendix S1).

## (c) Statistical regression modelling

To test the prediction that genetic variation and exchange in polar bears decreased along with declining sea ice coverage in the Svalbard Archipelago, we built linear mixed models for

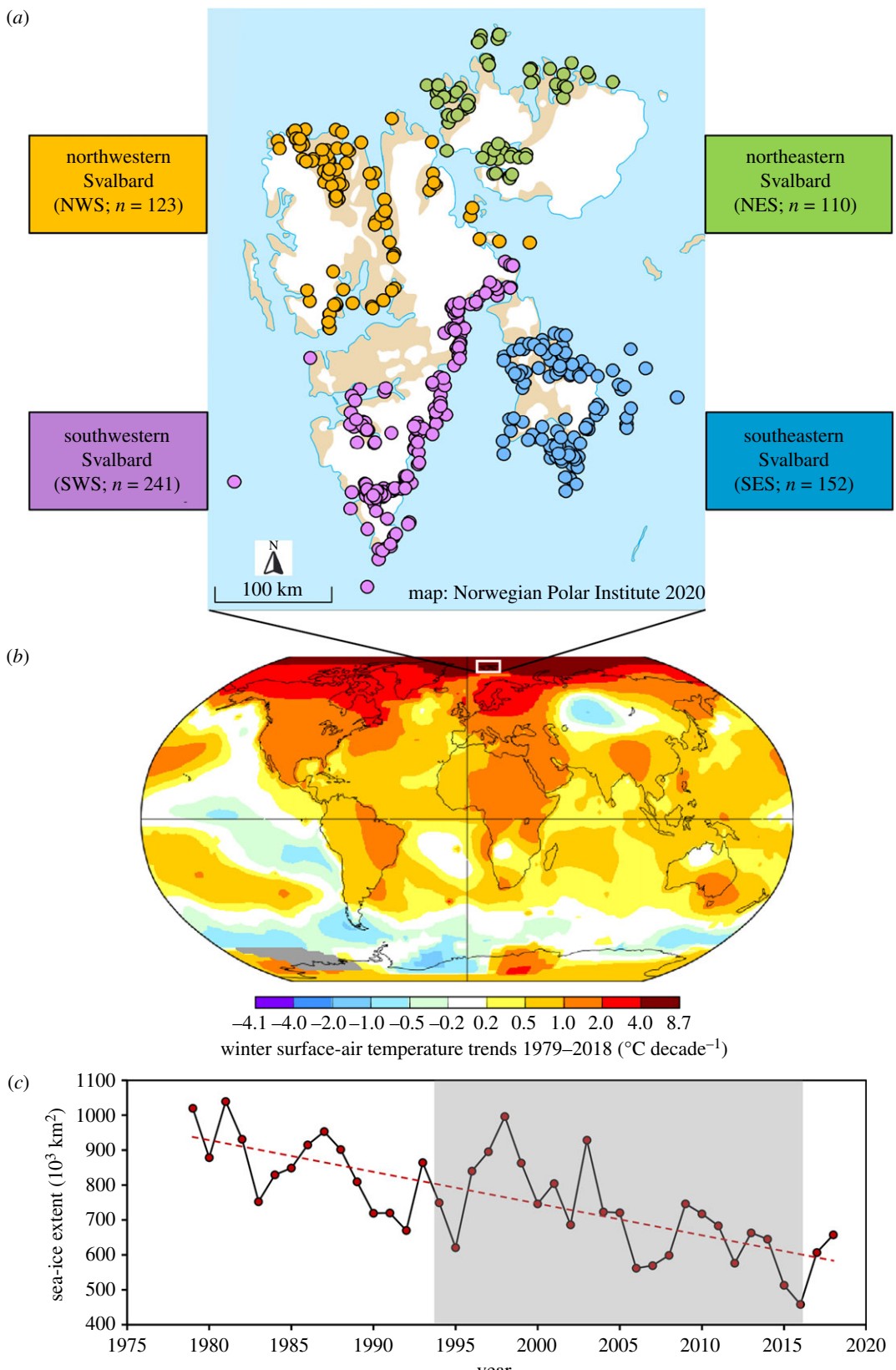

**Figure 1.** (*a*) Distribution of polar bears sampled in the present study across four geographical areas of the Svalbard Archipelago with sample sizes indicated. (*b*) Global surface-air temperature trends since 1979 (linear trends in °C decade$^{-1}$ for December–February). Data source: http://data.giss.nasa.gov/gistemp. (*c*) Sea ice extent trends in the Barents Sea for April (typical month with the highest prevalence of ice in the region) from 1979 to 2018. Data source: http://www.mosj.no. The period of the present study is shaded in grey (*c*). (Online version in colour.)

each genetic diversity parameter ($H_O$, $H_E$, $A_R$ and $r_w$) and for pairwise genetic differentiation $G''_{ST}$ using the R package *glmmTMB* v. 1.0.2.1 [34]. This was done using either sampling area or pair of sampling areas, period (highly correlated with the decline in mean sea ice extent per study period; $R^2 = 0.916$, $p = 0.0106$), and their interaction as explanatory variables.

Additionally, we used the four sampling areas (or, in the case of between area $G''_{ST}$, the six possible pairs of sampling areas) as random intercepts to account for temporal pseudoreplication resulting from estimating effects within and between the same areas (or pairs of areas) at different time points. For response variables that assume values between 0 and 1 ($H_O$, $H_E$ and

$G''_{ST}$), we used beta mixed regression analysis to fit models. For $A_R$ and $r_w$, we fitted models using linear mixed regression analysis. Owing to the properties of our dataset (samples size = 16), we selected the optimal model based on Akaike's information criterion corrected for small sample size (AIC$_C$) and Akaike weights ($\omega_i$) [35] (electronic supplementary material, appendix S1).

### (d) Spatio-temporal patterns of genetic clustering

To track potential changes in the composition and number of genetic subpopulations or clusters ($K$) of polar bears along with declining sea ice cover in the Svalbard Archipelago, we ran STRUCTURE v. 2.3.4 [36] for (i) all sampling areas and periods, (ii) per sampling area across periods (temporal component) and (iii) per period across sampling areas (spatial component). This was done both with and without the LOCPRIOR model, which uses additional sample-specific data, such as sampling location or time, to increase the power to detect subtle genetic changes and genetic structure. We also included runs accounting for kin structure following the recommendations by Waples & Anderson [37]. This was done to assess if the presence of family groups and cubs captured together with their mother in our data potentially affected the patterns observed. We applied the approach of Puechmaille *et al.* [38] to infer the optimal $K$ (electronic supplementary material, appendix S1).

### (e) Forward-in-time simulations and genetic bottlenecks analysis

To explore the potential long-term effects of sea ice loss on genetic diversity ($A_N$, $H_O$ and $H_E$) and global genetic differentiation ($G_{ST}$; Nei & Chesser [39]) in the four polar bear sampling areas in Svalbard, we first conducted individual-based forward-in-time simulations in NEMO-AGE v. 0.29.0 [40] (electronic supplementary material, appendix S1). We implemented a simplified demographic model of polar bears to account for generation time, number of adults and offspring, and the 3-year reproductive cycle of females [41] (figure 5$a$; detailed in the electronic supplementary material, appendix S1). Under fixed parameters, we compared six alternative scenarios: (1) no effect of sea ice loss on population connectivity, (2) a moderate effect of loss in sea ice on population connectivity, (3) pronounced effect of loss in sea ice on population connectivity, (4) pronounced effect of sea ice loss on population connectivity (with symmetrical gene flow rates) and size, (5) pronounced effect of sea ice loss on population connectivity (with asymmetrical gene flow rates between northern and southern areas at generation 0 to complete isolation by generation 10) and size, and (6) empirical genotypic data from the four capture areas at the last sampling period (T4 or T5) (figure 5; electronic supplementary material, appendix S1).

Finally, to test for recent bottlenecks, which would have resulted from a range contraction owing to ongoing climate change, we employed the heterozygosity excess and $M$-ratio tests implemented in INEST v. 2.2 [42] (electronic supplementary material, appendix S1). Lastly, we inferred historic bottlenecks (up to 100 generations back) using coalescent analysis as implemented in the R-package *DIYABCskylineplot* v. 1.0.1 [43,44] (electronic supplementary material, appendix S1).

## 3. Results

### (a) Temporal variation in genetic diversity, inbreeding and differentiation

Our results are consistent with the predicted erosion of genetic diversity locally during a time of sea ice loss. Model selection based on AIC$_C$ invariably identified time (i.e.

sampling period) as the main predictor variable needed to explain the observed variation in genetic diversity parameters. Over time, $A_R$ decreased by −0.145 (97.5% CI: −0.211 to −0.079) and $H_E$ by −0.041 (97.5% CI: −0.069 to −0.013; on the logit scale) per 5-year period, corresponding to approximately 3% and 9% loss through the whole study period, respectively. Furthermore, $r_w$ increased by 0.030 (97.5% CI: 0.014–0.044) per 5-year period (figure 2; electronic supplementary material, tables S2-1–S2-3). Moreover, when we used other plausible temporal divisions, we found that the pattern did not change (results not shown).

Although less supported by AIC$_C$, models including both study area and time as predictor variables provided additional insight. Specifically, they showed considerably lower genetic diversity and higher relatedness among polar bears from NWS, where sea ice loss has been particularly severe, compared to bears from the rest of the Svalbard Archipelago (figure 2; electronic supplementary material, tables S2–S4). Likewise, models using sampling period as a categorical rather than continuous variable resulted in lower support by AIC$_C$ (electronic supplementary material, tables S2–S5). Still, they provided evidence of the above-described temporal changes, while simultaneously pinpointing some potential deviations from the straight-line relationship of the model preferred by AIC$_C$.

Our results are also consistent with the predicted decline in genetic exchange locally following loss of sea ice coverage. Notably, pairwise $G''_{ST}$ between sampling areas increased by 0.512 (97.5% CI: 0.251–0.773) on the logit scale over time, and this increase was strongest in the last period and for the northern sampling areas (NWS and NES) (figure 3). A similar but non-significant pattern was observed when modelling temporal samples per area (0.311, 97.5% CI: −0.279–0.901). As observed for the genetic diversity indices, models of $G''_{ST}$ including also (pairwise) sampling area as predictor variable (or period as a factorial variable) were less supported by AIC$_C$ but provided additional predictive power by identifying significantly elevated genetic differentiation among the polar bears in the northern areas of Svalbard (figure 3). A sign test rejected the hypothesis of random fluctuations of genetic differentiation with $p = 0.022$ (increase in differentiation in 11 of 13 time steps).

### (b) Spatio-temporal patterns of genetic clustering

Our results provided evidence of increasing genetic structure coinciding with the loss of sea ice habitat. On average, across the whole dataset, the STRUCTURE results consistently identified $K = 3$ genetic clusters as the most likely model of spatial population structure (electronic supplementary material, figures S2–S4), suggesting a split among polar bear bears from NWS, SWS and eastern Svalbard (NES and SES) (figure 4). However, it was evident that the degree of admixture was initially high, declined over time and, eventually, sampling areas became unique genetic clusters, a pattern detected in NWS and SWS, and to some extent in SES. While STRUCTURE runs within each sampling area showed no clear separation between temporal samples (results not shown), STRUCTURE runs within each sampling period showed a pattern of increasing genetic heterogeneity over time (electronic supplementary material, figures S2–S5).

In alternative models of spatial genetic structure (figure 4), $K = 2$ showed a strong split between NWS and the rest of the sampling areas. At $K = 4$, a fourth cluster appeared in

Proc. R. Soc. B 288: 20211741

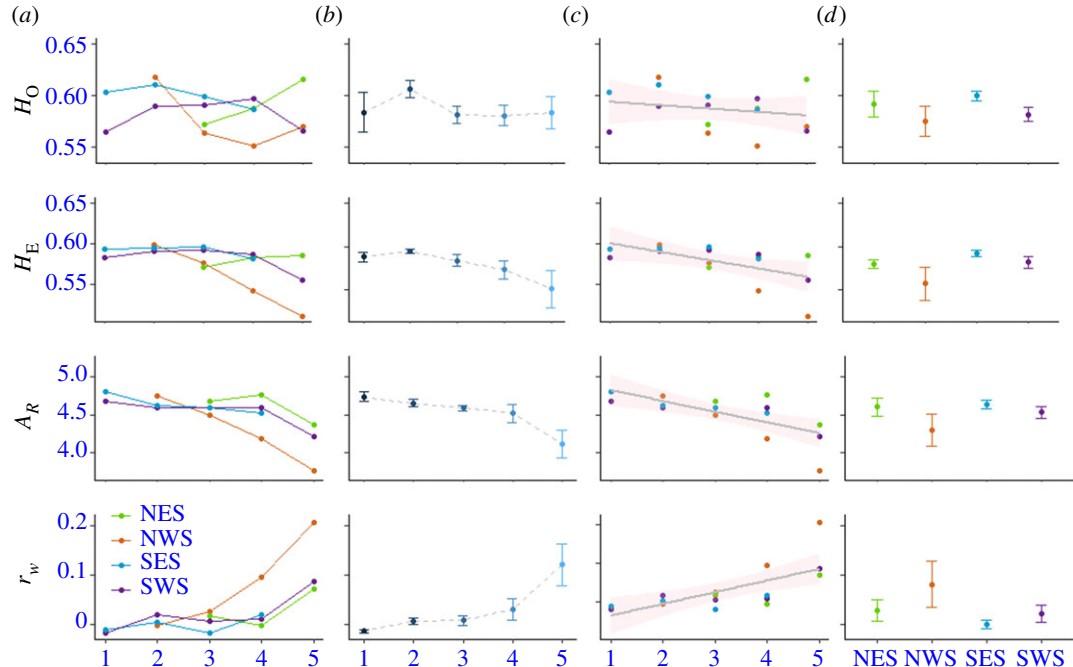

**Figure 2.** (a–c) Associations between period (correlated with sea ice loss) and mean observed ($H_O$) and unbiased expected heterozygosity ($H_E$) allelic richness ($A_R$) and coefficient of relatedness ($r_w$) in the four Svalbard polar bear sampling areas: (a) area x time interaction plot, (b) temporal effects as a factor where error bars represent $\pm$ s.e., and (c) temporal effects as continuous variable where the solid line depicts the overall regression trend for each response variable and shaded area around line depicts the 95% CI. (d) Within-area genetic diversity summary statistics ($\pm$s.e.) depicting area effects. (Online version in colour.)

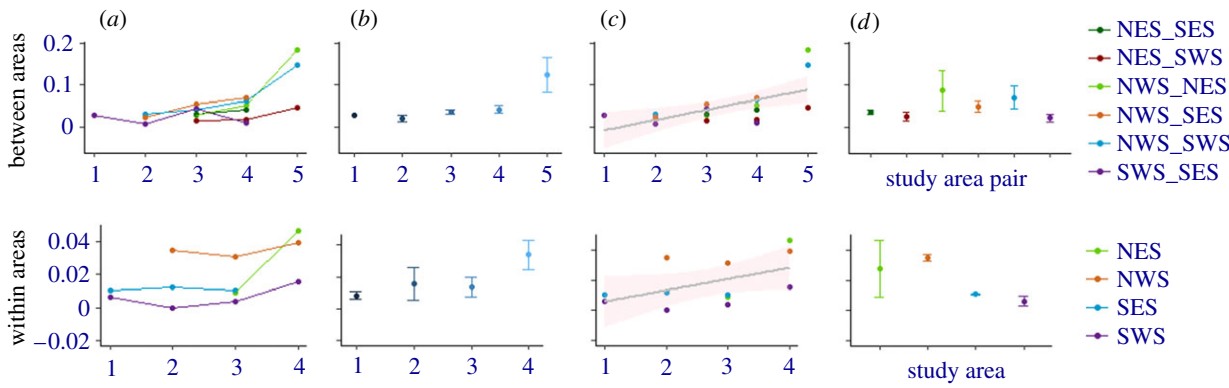

**Figure 3.** Within- and between-area trends of genetic differentiation (fixation index $G''_{ST}$). (a) pair x period (correlated with sea ice loss) interaction plot. (b) Association between period and $G''_{ST}$ between pairs of sampling areas for polar bears in Svalbard. The solid line depicts the overall regression trend for each response variable and shaded areas around the regression line depicts the 95% CI. (c) Area and among area effects of sea ice loss on $G''_{ST}$. (d) Within-area genetic diversity summary statistics ($\pm$s.e.) depicting area effects. (Online version in colour.)

southern Svalbard (SWS and SES), which was dominated by polar bears in SES and varied across periods. Likewise, a fifth cluster emerged at K = 5 in southern Svalbard, that was dominated by polar bears in SWS and was also temporally variable. When we accounted for kin structure by randomly purging closely related individuals, we obtained a similar clustering pattern (results not shown). As expected, STRUCTURE runs without LOCPRIOR gave fairly flat distributions of membership probabilities of individuals, which were resolved when using the LOCPRIOR model ($r_{\text{locprior}} = 2.99$).

## (c) Forward-in-time simulations and genetic bottlenecks

Our forward-in-time simulations indicated that the observed genetic erosion is a trend that will probably continue for future generations of the Barents Sea subpopulation. Scenarios 1 and 2 with no loss in sea ice and functional connectivity between populations was, as expected, the best for preserving

genetic diversity for the next 100 generations (figure 5). By contrast, scenarios with restricted gene flow (scenario 3) and diminishing gene flow over generations along with exponentially declining populations (scenarios 4 and 5) revealed that climate-induced habitat fragmentation owing to sea ice loss over the next 100 generations would result in further genetic erosion and increase in genetic differentiation (figure 5). Moreover, the simulations with the empirical genetic data (scenario 6) of patches already experiencing intensified genetic drift showed a sharper decline in genetic diversity and connectivity over the next 100 generations.

There was no support for recent genetic bottlenecks based on heterozygosity-excess or M-ratio tests, regardless of the parameter combination used ($p > 0.05$). The M-ratio values ranged from 0.726 to 0.891 across sampling areas per period (electronic supplementary material, tables S2–S7), while M-ratio values of less than 0.7 are generally considered as indicative of a bottleneck. The coalescent analysis indicated a rather constant historical effective population size,

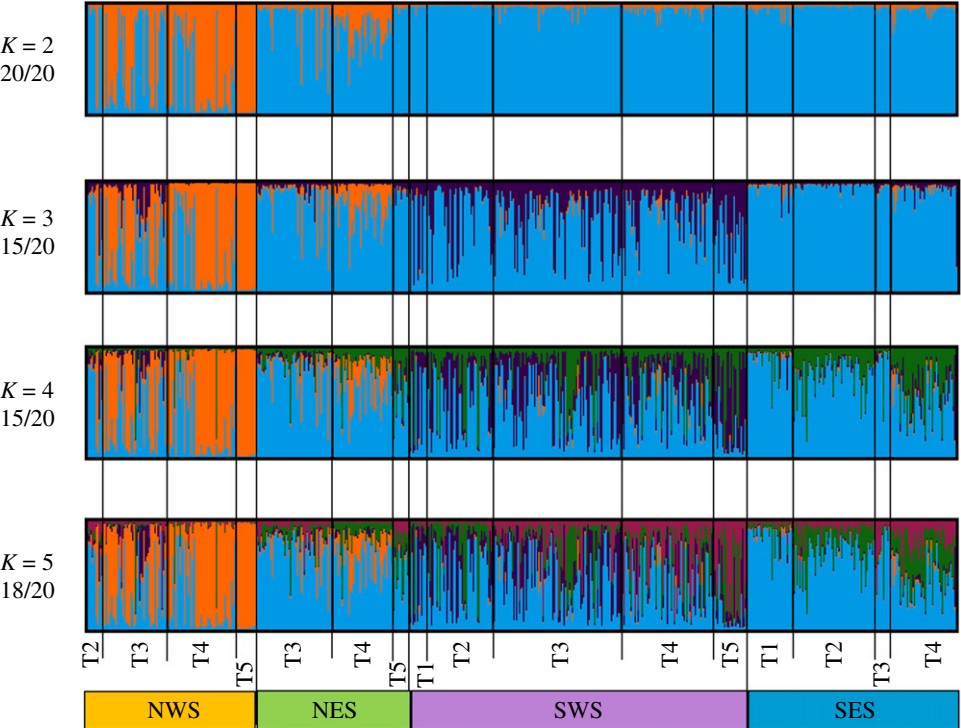

**Figure 4.** CLUMPAK-averaged Bayesian Sᴛʀᴜᴄᴛᴜʀᴇ clustering outputs for 20 independent runs of $K = 2$–5 for polar bears in Svalbard using spatio-temporal group as a prior. Numbers below each $K$-value show the proportion of runs that converged to the solution presented. Temporal points: T1 = 1995–1999; T2 = 2000–2004; T3 = 2005–2009; T4 = 2010–2014; T5 = 2015–2016. (Online version in colour.)

corroborating the findings of the heterozygosity-excess and $M$-ratio tests (electronic supplementary material, figures S2–S6).

## 4. Discussion

### (a) Climate-induced loss of genetic diversity

The disappearance of Arctic sea ice is accelerating and rapidly changing the habitat configuration and dispersal patterns of ice-dependent species, which, in turn, escalates conservation challenges in populations of threatened species across the Arctic [30,45]. The present study on polar bears of the Svalbard Archipelago reports the local loss of genetic diversity and exchange across four sampling areas over roughly 20 years that were characterized by substantial loss of sea ice coverage coinciding with major demographic changes [22,23]. Over time, such erosion of genetic diversity may reduce the fitness of individuals and cause an elevated risk of extinction [19,46]. The magnitude and rate of loss of genetic diversity and gene flow that we observed is alarming considering that polar bears have historically shown relatively little genetic differentiation even on a global scale [47–49], but now are facing increasingly strong climatic selective pressure. The results of simulations suggested that further loss of sea ice will lead to the continued erosion of local genetic diversity in polar bears of the Svalbard Archipelago and to increased isolation between local areas, especially if there is a concurrent decrease in the number of bears.

### (b) Rapid genetic differentiation correlates with sea ice loss

For polar bears, sea ice is the intervening habitat matrix that either promotes or restricts functional connectivity among habitats and subpopulations [17,50]. In the polar basin,

global ecoregions are defined by the presence or absence of sea ice, and gene flow between polar bear subpopulations is shaped by glacial and open water barriers to dispersal. Migration between subpopulations and local genetic diversity are expected to decrease as open water barriers to dispersal are present for increasingly long seasons. Accordingly, genetic responses to sea ice loss are likely to be subpopulation-specific and variable in time and space [17,47,51,52].

Our null hypothesis of panmixia of polar bears across the Svalbard Archipelago was rejected owing to major temporal changes in genetic composition during the course of the focal time period. Initially, the four assigned sampling areas showed a high degree of admixture, which decreased over time and covaried with documented reductions in sea ice coverage [20]. At the end of the study period, the four assigned sampling areas were genetically differentiated because of intensified genetic drift and probably also elevated site fidelity owing to the decreasing number of pelagic bears. Studies have shown that female bears in the Svalbard Archipelago and in the Beaufort Sea are highly philopatric [33,53,54]. In fact, philopatry to summering areas is common in bears that experience reductions in summer ice cover [55–59]. These behavioural traits (site fidelity and sex-biased philopatry) also act as biological drivers of genetic differentiation among areas in bears [51,60–62]. The genetic signature of behavioural traits could be masked by high gene flow (homogenizing force), which counteracts the effects of genetic drift (diversifying force). Indeed, during periods with extensive sea ice, migration and gene flow through predominantly male 'pelagic' polar bears over a larger geographical scale in Svalbard is likely [63]. Our findings suggested that gene flow mediated by 'pelagic' polar bears, which connect the various areas of Svalbard during the mating season, has been gradually declining owing to the loss of sea ice.

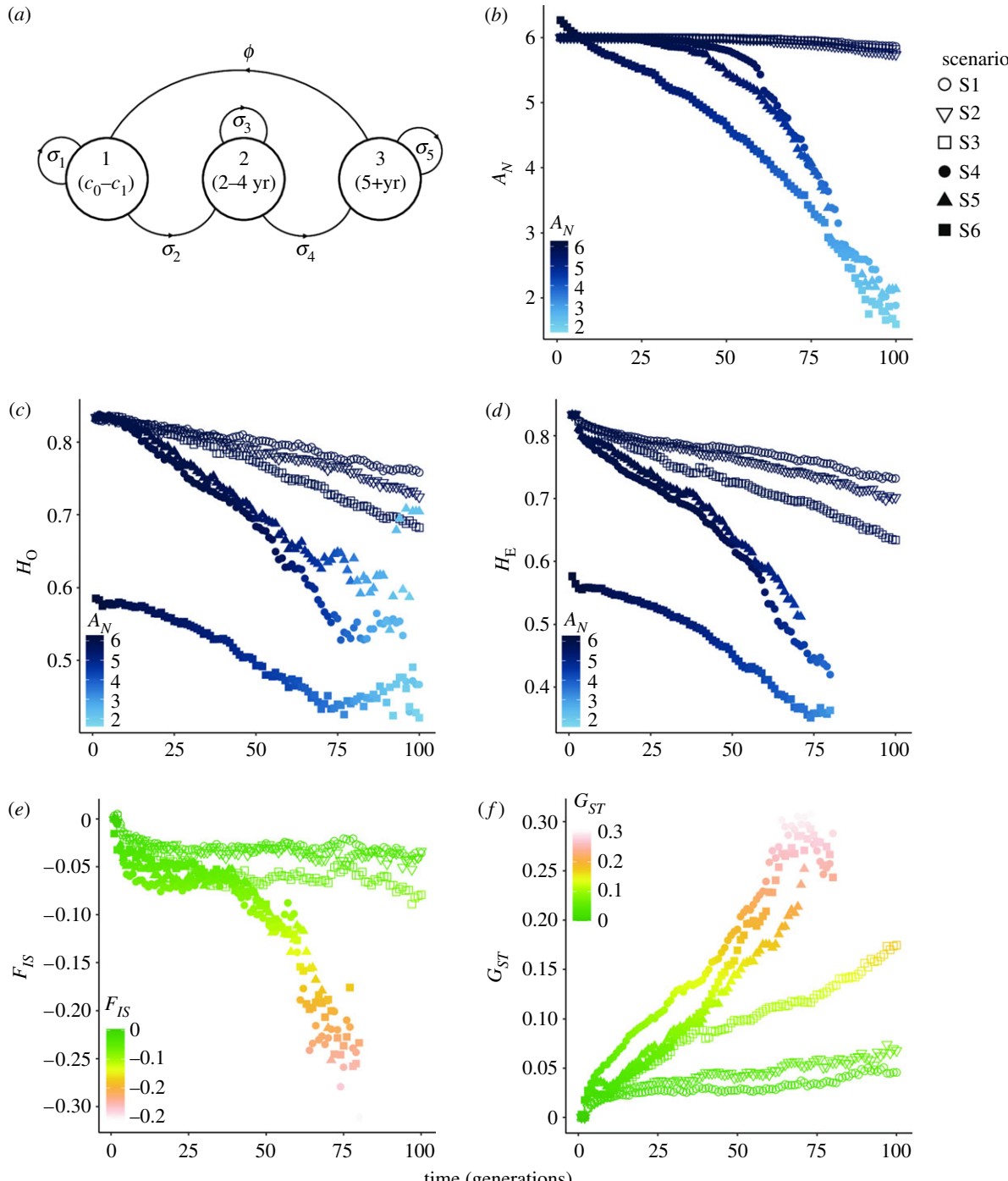

**Figure 5.** Forward-in-time simulations of the genetic response to climate-driven sea ice reduction and possible polar bear population scenarios. (*a*) The simplified life-history graph of polar bears used for the simulations illustrating a three-stage life cycle: cubs (cub-of-the-year (C0) and yearling (C1); stage 1)—subadults (stage 2)—adults (stage 3) where survival rates are denoted by $\sigma$. (*b,c*) temporal changes in genetic diversity ($A_N$ $H_0$ and $H_E$) over 100 generations under different scenarios where $H_0$ and $H_E$ are colour coded based on $A_N$. (*e,f*) temporal changes in genetic differentiation ($G_{ST}$; Nei & Chesser [39]) over 100 generations under different scenarios. Constant population size scenario with symmetrical gene flow rate of 50% (scenario 1) 25% (scenario 2) and 0.1% (scenario 3) between patches. Habitat fragmentation scenario with a 10-fold gene flow rate decline per 30 generations (scenario 4) and with asymmetrical gene flow rates between northern and southern patches (scenario 5) where patches were undergoing 97.5% exponential population decline. Scenario 5 is based on the empirical dataset (scenario 6). (Online version in colour.)

With our extensive spatio-temporal sampling regime, we were able to disentangle the effects of sea ice loss and behavioural traits. For instance, we determined that genetic drift was more intense in NWS, which is the area with the highest loss of sea ice and degree of philopatric bears. By contrast, the southern areas with relatively lower levels of sea ice loss showed relatively weaker signals of genetic differentiation, even though there was evidence of localized genetic groups indicative of site fidelity and/or philopatry. In turn, the local loss of genetic diversity may be best explained by a decrease in gene flow resulting in increased inbreeding of local polar bears within the focal sampling areas. In addition, restricted movements across relatively small geographical areas may also reflect sea ice affecting prey distribution [64,65].

Indeed, we found that the levels of admixture between individuals from the genetically differentiated sampling areas decreased over time. The observed temporal decline in admixture was especially pronounced between the northern and

southern range extremes characterized by different distinct climatic regimes in the Barents Sea [25]. Accordingly, we propose that the loss of sea ice is the ecological driver for the declining migration of 'pelagic' polar bears in the Svalbard Archipelago and a relative increase of the proportion of 'local' polar bears showing site fidelity. On a broader scale, our findings meet those of the Foxe Basin and Baffin Bay polar bear subpopulations [17,52], where the loss of sea ice establishes open oceanic waters as a barrier to gene flow. Moreover, in the absence of significant signals of heterozygosity excess and changes in population size, the observed pattern of genetic structure of polar bears in the Svalbard Archipelago is likely to have appeared recently because of intensified genetic drift.

## (c) Conservation implications

Our study shows that tracking of temporal population-genetic changes will become important for our understanding of how polar bears and other ice-dependent species may cope with rapid habitat fragmentation and loss in the Arctic and Antarctic. For the Svalbard Archipelago, the high degree of philopatry of 'local' female polar bears over generations [33,63] in combination with an expected further reduction in gene flow from the 'pelagic' part of the Barents Sea subpopulation render a significant future loss of genetic variation and, thus, increased population structure, likely. Although studies have reported a low level of inbreeding for polar bears in the Svalbard Archipelago [33]; increased levels of isolation between populations may increase inbreeding in the future, most likely with negative effects such as inbreeding depression. The NWS area may already today indicate future developments. The west coast of Spitsbergen has experienced the greatest loss of sea ice in the Barents Sea region [25] and also showed the highest rate of change in genetic diversity in the current study. The three genetic clusters detected in our study indicate that even on a relatively small geographical scale, distinct management units of polar bears have already evolved in the Svalbard Archipelago. Furthermore, in line with the findings of Aars et al. [29], our study suggests that northeastern Svalbard may already

represent a refugia for the species. Accordingly, genetic monitoring of polar bears for successive generations in Svalbard is warranted and highly recommended in order to follow the detected declining trend of genetic diversity owing to the predicted loss of sea ice in the region.

Ethics. The Norwegian Polar Institute, Tromsø, Norway, collected the samples used in this study during a long-term monitoring project on the ecology of polar bears in the Barents sea. Polar bear captures were carried out in accordance with guidelines and regulations from the Governor of Svalbard and were approved by the Norwegian Animal Research Authority (PO Box 8147 Dep., N-0033 Oslo, Norway).

Data accessibility. The authors declare that data supporting the findings of this study are available within the article and its supplementary information files [66]. The genetic dataset and relevant metadata are available from the Dryad Digital Repository: https://doi.org/10.5061/dryad.zpc866t8x [67].

Authors' contributions. S.N.M.: conceptualization, data curation, formal analysis, methodology, visualization, writing—original draft, writing—review and editing; J.A.: conceptualization, data curation, funding acquisition, investigation, resources, writing—original draft, writing—review and editing; I.F.: data curation, investigation, methodology, validation, writing—review and editing; C.F.C.K.: conceptualization, writing—review and editing; E.M.L.Z.F.: data curation, investigation, writing—review and editing; Ø.W.: investigation, resources, writing—review and editing; D.E.: formal analysis, investigation, methodology, validation, visualization, writing—original draft, writing—review and editing; M.A.: investigation, writing—review and editing; L.B.: resources, writing—review and editing; A.E.D.: resources, writing—review and editing; T.N.: formal analysis, validation, visualization, writing—review and editing; H.G.E.: conceptualization, supervision, writing—original draft, writing—review and editing; S.B.H.: conceptualization, formal analysis, funding acquisition, project administration, supervision, validation, visualization, writing—original draft, writing—review and editing. All authors gave final approval for publication and agreed to be held accountable for the work performed therein.

Competing interests. The authors declare no competing interests.

Funding. The World Wildlife Fund supported the fieldwork with significant funding. S.N.M. is grateful to the Norwegian Institute of Bioeconomy Research for financial support.

Acknowledgements. The Airlift helicopter company provided high-quality service to the Norwegian Polar Institute during fieldwork. Statoil supported the program from 2006 to 2009.

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
