## [Peer Review File · Proceedings of the Royal Society B: Biological Sciences]

Review History

RSPB-2021-0011.R0 (Original submission)

Review form: Reviewer 1

Recommendation

Accept with minor revision (please list in comments)

Scientific importance: Is the manuscript an original and important contribution to its field?

Good

General interest: Is the paper of sufficient general interest?

Acceptable

Quality of the paper: Is the overall quality of the paper suitable?

Good

Is the length of the paper justified?

Yes

Should the paper be seen by a specialist statistical reviewer?

No

Do you have any concerns about statistical analyses in this paper? If so, please specify them explicitly in your report.

Yes

It is a condition of publication that authors make their supporting data, code and materials available - either as supplementary material or hosted in an external repository. Please rate, if applicable, the supporting data on the following criteria.

Is it accessible?

Yes

Is it clear?

N/A

Is it adequate?

N/A

Do you have any ethical concerns with this paper?

No

Comments to the Author

I appreciate the opportunity to review this manuscript. Other than the technical genetic material in Methods, which needs to be made more accessible for non-specialists, the manuscript was well written and easy to read and understand. I found the most compelling aspect of the manuscript to be the spatial and temporal changes in diversity, and some of my specific comments below suggest that the practical significance of those results might be better explained and given more prominence.

1. Lines 133-139. The title of Table S1 includes the number of individuals in each of three age classes. The inclusion of bears ≤ 2 years old makes me wonder whether data from family groups were included and, if so, how the dependence (shared genetics) between individuals could be expected to influence the results. Similarly, were data from individuals observed in more than one year included and, if so, how might that influence the results? It would be helpful to describe the data more fully, including such aspects as mentioned above, in the Methods.
2. Lines 164-184. The text in these two sections is probably only fully understandable to a population geneticist. Please consider your non-geneticist readers and add some detail about what these statistics measure and how their use contributes to the objectives of the study. Also, it is probably important to understand whether the various statistics are computed on the basis of an individual bear, a pair of bears, or a sample of bears. If other than a sample of bears, how is the information generalized to a sampling area and period?
3. Lines 180-181. What are the permutations based on; permutations of what?
4. Lines 194-196. It is unwise to simply select the model with lowest AIC. For example, models with slightly larger AIC having fewer parameters would generally be preferred.
5. Lines 197-198. Why are differences between the AICs of competing models important? The text in the preceding paragraph states that the model with the smallest AIC was selected as the preferred model, with no mention of model comparison. Does this sentence apply to the mixed models that seem to be the main subject of this paragraph? If so, consider moving this sentence to later in the paragraph and adding an explanation of how model comparisons contributed to the analysis.

6. Lines 201-207. Please clarify this section of text. For example, if measuring differentiation between the same four areas, why would levels of differentiation between areas change? Is the differentiation between two areas not fixed? Also, how are the reduced data sets being formed? What is the larger collection of data from which subsets are being selected?
7. Lines 212-218. Please expand this section so that a non-geneticist can understand it. What is the LOCPRIOR model? The three-level hierarchy is not clear. Why is it necessary to account for kin structure? What is K?
8. Line 220-231. The first sentence of this paragraph is an example of how the Methods could be improved throughout; most readers will understand a genetic bottleneck and that relatively non-technical information is used to motivate the tests, which most readers will otherwise not understand. Please add more of this sort of information to this section and the preceding sections to improve the understanding of general readers. For clarity, is the range contraction of subpopulations and reduced gene flow between subpopulations what could result in a bottleneck? Is subpopulation synonymous with sampling area?
9. Line 233. What is NA? I don't think that quantity has been introduced.
10. Line 236. Was sex stochastically determined?
11. Line 237. A Poisson distribution with mean 2 generates an outcome equal to 0 or exceeding 3 with probabilities of approximately 0.14 (each). That is not a suitable model for polar bears. Most subpopulations of polar bears have either 1 or 2 offspring, with triplets being rare. It would seem that having too many large litters would substantially hasten the loss of genetic diversity; is that correct? How useful are these simulation results when the reproductive model is so unrealistic?
12. Line 240. How do the reproductive model (Poisson mean 2) and the population size model (constant or exponential decline) coexist?
13. Line 241. Is my interpretation that $m = 0.5$ means that individual bears have a 50% probability of moving to a different patch each year correct? If so, how was the population size held constant with what appears to be stochastic migration between patches (similar to the preceding question)? Why was this value chosen; is it a realistic value for the population at this point in time?
14. Line 243. Is there any rationale for this pattern in the decline in migration rates, such as somehow linking m to projections of sea ice loss?
15. Lines 243-244. Please clarify whether g_i refers to generation i . A intermediate model in which a low level of migration persisted might have been interesting. Is there any realistic expectation that gene flow might not reach 0 even if sea ice disappears? For example, do (might) bears swim between areas?
16. Line 246. Is the 0.975 the λ parameter in an exponential decay model? Is this an annual or generational rate? What is the expected number of bears left after 120 years? Is the decay stochastic or deterministic?
17. Lines 249-252. Please clarify the fourth model. In particular, how was the empirical genetic data used to parameterize the model? Also, are the rates of gene flow constant from g_0 to g_5 ?
18. Lines 263-264. Consider reorganizing the supplementary material so that the table numbers in the two supplements are numbered differently (e.g., both now have a Table S1).

Maybe something like S1-1 and S2-1?).

19. Lines 265-266. Was weighted linear regression mentioned in Methods? Is linear regression or weighted linear regression most suitable for this problem? Do they provide fundamentally different information so that both are needed?

20. Line 284. I'm not confident that I understand what the statistics in Table S4 measure, but I'm wondering if it is meaningful to compare two different areas in two different time periods. For example, what does a comparison of NWS_T5 and SWS_T1 reveal?

21. Line 332. I think the reference should be to Table S5.

22. Line 334. In Figure S7, why does the effective population size increase in the last few generations and the "95% highest posterior density estimates" get so much wider?

23. Lines 385-386. Does "significantly genetically differentiated" refers to a statistical significance? If so, consider also addressing practical significance. Are the observed differences meaningful in a biological/ecological context? For example, how do the differences between sampling areas compare within the scope of subpopulation differentiation on a global scale?

24. Line 412. Is the text "conservation implications" the conclusion of the sentence or a heading (as it currently appears)?

25. Lines 430-431. Does the idea that distinct management units are emerging depend on the practical significance of the genetic differentiation between sampling areas (see earlier comment about statistical vs. biological significance)?

26. Line 433. I'll make a comment here with respect to "crucial refugia", but it could be made almost anywhere in the Discussion. An aspect that seems to be lacking from the Discussion is a consideration of the likelihood that bears will emigrate to more favorable habitat. I suppose that could still be portrayed as a loss of diversity from Svalbard, although those genes would persist elsewhere.

Review form: Reviewer 2

Recommendation

Major revision is needed (please make suggestions in comments)

Scientific importance: Is the manuscript an original and important contribution to its field?

Good

General interest: Is the paper of sufficient general interest?

Good

Quality of the paper: Is the overall quality of the paper suitable?

Good

Is the length of the paper justified?

Yes

Should the paper be seen by a specialist statistical reviewer?

No

Do you have any concerns about statistical analyses in this paper? If so, please specify them explicitly in your report.

Yes

It is a condition of publication that authors make their supporting data, code and materials available - either as supplementary material or hosted in an external repository. Please rate, if applicable, the supporting data on the following criteria.

Is it accessible?

Yes

Is it clear?

Yes

Is it adequate?

Yes

Do you have any ethical concerns with this paper?

No

Comments to the Author

I found the paper interesting, well-written, and a unique way to use a long-term data set. I was a bit surprised to see that there was significant genetic differentiation over such a (relatively) small area but you did a good job of explaining why the behavioral differences of bears in the sub-population could lead to the observed patterns. I do have some concerns with the statistical tests though as it does not appear that the methods used are appropriate given the range of values possible for most of the dependent variables analyzed (see below for more details). As with so much in statistics, it might be that results derived from more appropriate statistical tests do not differ significantly from the approach originally taken. However, I think it is still important to analyze the data with the correct models to make sure that results are just an artifact of inappropriate models.

One of my biggest comments is to put into better perspective the biological implications of this level of loss of genetic diversity. For example, in your abstract you state that you found "slight but significant genetic differentiation" but throughout the manuscript I didn't really see much discussion about what this level of genetic differentiation really means biologically. Is it really significant enough to lead to negative population-level consequences greater than the effects from causative agent, i.e., sea ice loss? I think this last point deserves a fair amount of discussion to better present the results of this study. I just wonder if loss of genetic diversity is really overly concerning given that polar bears generally don't exhibit that much genetic diversity, globally, to begin with compared to other species and ursids. Given that the same factor that leads to loss in gene flow also has been shown to lead to reduced survival and productivity, won't those have a greater and sooner impact on populations than reduced gene flow and genetic diversity? It just seems like the time scale for genetic effects to manifest themselves into negative population-level consequences is much longer than the direct effects of sea ice loss leading to longer periods fasting on land, reduced access to prey, etc. Again, I'm not discounting the importance of considering the consequences of sea ice loss on polar bear genetics, but would just like it to be put into better context with other factors that are likely to be stronger drivers of population dynamics related to sea ice loss.

Specific comments:

L75: Delete "is not"

L79-80: I am not disputing your results, but I think it is worth further discussion the fact that both the pelagic and "local" bears continue to overlap during the mating season thus calling into question the mechanism presented in the rest of the paper for why there is genetic differentiation

occurring. It shouldn't matter if the two groups are isolated during other periods of the year outside of the breeding season. Anyway, some additional discussion about why differentiation is occurring even when there is overlap during the breeding season would be valuable.

L189-190: I think it would be more appropriate to treat period as a categorical variables. That comports more directly with the structure of the data. Is there a reason for treating it as continuous? If you continue to use it as a continuous variable, it would be good to see that it doesn't lead to different results if treating it as categorical variable.

L192-193: Please provide results of the correlation between the period category and sea ice extent.

L194: I don't think your use of a simple linear model is appropriate for the analysis of these dependent variables. Each of these dependent variables does not have the proper support to apply a simple linear model without at least some sort of transformation. A simple linear model would require the dependent variables to be able to take values >0 and <0 and none do. Additionally, some of the variables can only take values that occur between 0-1 necessitating the use of a Beta regression or something else that supports variables that can only range between 0 and 1.

L199-200: Similar to the above comment, I don't think a linear mixed model is the appropriate test for this dependent variable given that its support ranges from 0-1. This needs a Beta-type regression too (or something similar) with a random intercept. And the square-root transformation described below (L207) does not help with the fact that a linear model will allow negative values in the dependent variable even though they are not possible.

L237: While the Poisson distribution is relevant for simulating discrete values as you'd expect for litter sizes, I don't believe it is being appropriately applied here. Drawing from a Poisson distribution with lambda equal to 2 can leading to litter sizes being drawn that are impossible. It actually leads to ~14% of litters being simulated with litter sizes >3 and ~14% with 0 cubs. A better approach would be a multinomial distribution where probabilities for each of 3 litter sizes (1,2,3) would be applied. This forward-in-time simulation also seems a little simplistic. For example, it appears that bears are allowed to mate and reproduce annually, even though there is at least a 2-3 year time-lag between when females re-breed after having a litter. It doesn't appear that this is accommodated currently.

L241-251: I'd like to see some additional text describing what led to these choices in parameter values for the model.

L255-259: Setting aside the issues I have with the choice of model, while AICc scores helped find the model (in your set) with the most support, it does not saying anything about how predictive the model is. Did you perform any sort of analysis (e.g., cross validation) to assess the predictability of the model and not just the fit compared to other models in the set tested?

Review form: Reviewer 3

Recommendation

Accept with minor revision (please list in comments)

Scientific importance: Is the manuscript an original and important contribution to its field?

Excellent

General interest: Is the paper of sufficient general interest?

Excellent

Quality of the paper: Is the overall quality of the paper suitable?

Good

Is the length of the paper justified?

Yes

Should the paper be seen by a specialist statistical reviewer?

No

Do you have any concerns about statistical analyses in this paper? If so, please specify them explicitly in your report.

No

It is a condition of publication that authors make their supporting data, code and materials available - either as supplementary material or hosted in an external repository. Please rate, if applicable, the supporting data on the following criteria.

Is it accessible?

Yes

Is it clear?

Yes

Is it adequate?

Yes

Do you have any ethical concerns with this paper?

No

Comments to the Author

This study analyzed the spatiotemporal genetic variation of polar bears in the Svalbard archipelago and its association with sea ice decline over a period of 22 years. I found the data properly analyzed, the paper well written, and the conclusions justified. This paper provides new and significant insights into the impact of climate change on polar bears and potentially other sea ice dependent species.

The only main comment I have concerns the discussion of the population genetic differentiation. I find the authors possibly overstate (e.g., p. 20, line 383-386), or incompletely discuss (e.g., p. 20, line 401-403), the genetic structure among Svalbard polar bears.

First of all, the genotyped polar bear individuals were divided into four geographical areas. The appendix refers to previous studies of movement ecology and sea ice presence as the reason for this geographic division, but because it makes an important basis for the analyses and because STRUCTURE analysis seems to favor 3 clusters (and not 4) and it shows significant gene flow among at least some areas for some or all the time periods, the background for these four areas could be explained more and better justified in the main text.

The STRUCTURE analysis also seems to recognize significant variability within the prior designated geographic clusters (Fig. 4 and also seen in Fig. S6), suggesting considerable gene flow and little genetic structure across most of the archipelago. I am not convinced this study has sufficient resolution to see that "The four assigned sampling areas were significantly genetically differentiated, indicating restricted gene flow amongst areas" (p. 20, line 385-386), e.g., there appear to be significant gene flow at least among SWS and SES, and again, just 3 clusters were supported by STRUCTURE. This could be discussed in more detail, or the discussion could be

more balanced.

However, the increasing genetic differentiation over time, especially with increasing isolation of NWS from the rest, seems clear and is an important finding, and if possible, STRUCTURE plots for each of the time periods (similar to Fig. S6) could be included in the main text, or perhaps it could be made clearer from their Fig. 4 (e.g., see below comment about sorting of the individual assignment of clusters).

Some additional minor comments:

P. 4, line 74-75: This sentence needs to be corrected: "...when there is no continuous sea ice cover is not surrounding Svalbard..."

P. 7, line 139: It could be mentioned how potential overlap between individuals among spatiotemporal groups are checked (e.g., have mark-recapture been employed to exclude recaptured individuals?).

P. 8, line 161: For clarity, it could also be mentioned here that the dataset comprised 622 individuals (and not 626 as listed here).

P. 20, line 403: Although perhaps briefly mentioned later in the discussion (line 427-428), it would be worth including a little more detail here about how the distinction in climate regimes between northern and southern ranges can drive the genetic differentiation/reduced gene flow, as this is an important finding of this study.

P. 21, line 409-411: Parts of this sentence is missing or need rephrasing.

Figure 2: Please describe what the dots in the plots represent (medians/averages/other of the different summary stats?).

Figure 4: If sorting the fraction of ancestral population (assignment of individuals to clusters) within areas in the plots (e.g., low to high proportion of orange cluster in each time period of NWS), it may be easier to follow the variation in genetic structure within the designated areas and over time.

It looks like the microsatellite genotyping data is made available through the DRYAD Repository and I encourage the authors to make sure the data is in a format that can easily be interpreted and utilized by the community.

Decision letter (RSPB-2021-0011.R0)

01-Apr-2021

Dear Dr Maduna:

I am writing to inform you that your manuscript RSPB-2021-0011 entitled "Sea ice reduction drives genetic differentiation among Barents Sea polar bears" has, in its current form, been rejected for publication in Proceedings B.

This action has been taken on the advice of referees, who have recommended that substantial revisions are necessary. With this in mind we would be happy to consider a resubmission, provided the comments of the referees are fully addressed. However please note that this is not a provisional acceptance.

Sincerely,
Dr Daniel Costa
mailto:proceedingsb@royalsociety.org

Associate Editor
Board Member: 1
Comments to Author:

All three reviewers give positive assessments overall. However two raise a number of detailed concerns about the appropriateness of statistical analyses used and model selection (with the latter, a point is raised about what could be simplistic reliance on the lowest AIC value when selecting a model, something that I have wondered about more generally - what exactly to small differences in AIC mean when comparing different models?). As one reviewer notes, carrying out more appropriate statistical analyses may not end up changing the conclusions, but it is still important for it to be done. One reviewer highlights that parts of the text would benefit from being revised to make it more accessible to the more general reader, which I agree is important.

Reviewer(s)' Comments to Author:
Referee: 1
Comments to the Author(s)

I appreciate the opportunity to review this manuscript. Other than the technical genetic material in Methods, which needs to be made more accessible for non-specialists, the manuscript was well written and easy to read and understand. I found the most compelling aspect of the manuscript to be the spatial and temporal changes in diversity, and some of my specific comments below suggest that the practical significance of those results might be better explained and given more prominence.

1. Lines 133-139. The title of Table S1 includes the number of individuals in each of three age classes. The inclusion of bears ≤ 2 years old makes me wonder whether data from family groups were included and, if so, how the dependence (shared genetics) between individuals could be expected to influence the results. Similarly, were data from individuals observed in more than

one year included and, if so, how might that influence the results? It would be helpful to describe the data more fully, including such aspects as mentioned above, in the Methods.

2. Lines 164-184. The text in these two sections is probably only fully understandable to a population geneticist. Please consider your non-geneticist readers and add some detail about what these statistics measure and how their use contributes to the objectives of the study. Also, it is probably important to understand whether the various statistics are computed on the basis of an individual bear, a pair of bears, or a sample of bears. If other than a sample of bears, how is the information generalized to a sampling area and period?

3. Lines 180-181. What are the permutations based on; permutations of what?

4. Lines 194-196. It is unwise to simply select the model with lowest AIC. For example, models with slightly larger AIC having fewer parameters would generally be preferred.

5. Lines 197-198. Why are differences between the AICs of competing models important? The text in the preceding paragraph states that the model with the smallest AIC was selected as the preferred model, with no mention of model comparison. Does this sentence apply to the mixed models that seem to be the main subject of this paragraph? If so, consider moving this sentence to later in the paragraph and adding an explanation of how model comparisons contributed to the analysis.

6. Lines 201-207. Please clarify this section of text. For example, if measuring differentiation between the same four areas, why would levels of differentiation between areas change? Is the differentiation between two areas not fixed? Also, how are the reduced data sets being formed? What is the larger collection of data from which subsets are being selected?

7. Lines 212-218. Please expand this section so that a non-geneticist can understand it. What is the LOCPRIOR model? The three-level hierarchy is not clear. Why is it necessary to account for kin structure? What is K?

8. Line 220-231. The first sentence of this paragraph is an example of how the Methods could be improved throughout; most readers will understand a genetic bottleneck and that relatively non-technical information is used to motivate the tests, which most readers will otherwise not understand. Please add more of this sort of information to this section and the preceding sections to improve the understanding of general readers. For clarity, is the range contraction of subpopulations and reduced gene flow between subpopulations what could result in a bottleneck? Is subpopulation synonymous with sampling area?

9. Line 233. What is NA? I don't think that quantity has been introduced.

10. Line 236. Was sex stochastically determined?

11. Line 237. A Poisson distribution with mean 2 generates an outcome equal to 0 or exceeding 3 with probabilities of approximately 0.14 (each). That is not a suitable model for polar bears. Most subpopulations of polar bears have either 1 or 2 offspring, with triplets being rare. It would seem that having too many large litters would substantially hasten the loss of genetic diversity; is that correct? How useful are these simulation results when the reproductive model is so unrealistic?

12. Line 240. How do the reproductive model (Poisson mean 2) and the population size model (constant or exponential decline) coexist?

13. Line 241. Is my interpretation that $m = 0.5$ means that individual bears have a 50% probability of moving to a different patch each year correct? If so, how was the population size held constant with what appears to be stochastic migration between patches (similar to the preceding

question)? Why was this value chosen; is it a realistic value for the population at this point in time?

14. Line 243. Is there any rationale for this pattern in the decline in migration rates, such as somehow linking m to projections of sea ice loss?

15. Lines 243-244. Please clarify whether g_i refers to generation i . A intermediate model in which a low level of migration persisted might have been interesting. Is there any realistic expectation that gene flow might not reach 0 even if sea ice disappears? For example, do (might) bears swim between areas?

16. Line 246. Is the 0.975 the λ parameter in an exponential decay model? Is this an annual or generational rate? What is the expected number of bears left after 120 years? Is the decay stochastic or deterministic?

17. Lines 249-252. Please clarify the fourth model. In particular, how was the empirical genetic data used to parameterize the model? Also, are the rates of gene flow constant from g_0 to g_5 ?

18. Lines 263-264. Consider reorganizing the supplementary material so that the table numbers in the two supplements are numbered differently (e.g., both now have a Table S1. Maybe something like S1-1 and S2-1?).

19. Lines 265-266. Was weighted linear regression mentioned in Methods? Is linear regression or weighted linear regression most suitable for this problem? Do they provide fundamentally different information so that both are needed?

20. Line 284. I'm not confident that I understand what the statistics in Table S4 measure, but I'm wondering if it is meaningful to compare two different areas in two different time periods. For example, what does a comparison of NWS_T5 and SWS_T1 reveal?

21. Line 332. I think the reference should be to Table S5.

22. Line 334. In Figure S7, why does the effective population size increase in the last few generations and the "95% highest posterior density estimates" get so much wider?

23. Lines 385-386. Does "significantly genetically differentiated" refers to a statistical significance? If so, consider also addressing practical significance. Are the observed differences meaningful in a biological/ecological context? For example, how do the differences between sampling areas compare within the scope of subpopulation differentiation on a global scale?

24. Line 412. Is the text "conservation implications" the conclusion of the sentence or a heading (as it currently appears)?

25. Lines 430-431. Does the idea that distinct management units are emerging depend on the practical significance of the genetic differentiation between sampling areas (see earlier comment about statistical vs. biological significance)?

26. Line 433. I'll make a comment here with respect to "crucial refugia", but it could be made almost anywhere in the Discussion. An aspect that seems to be lacking from the Discussion is a consideration of the likelihood that bears will emigrate to more favorable habitat. I suppose that could still be portrayed as a loss of diversity from Svalbard, although those genes would persist elsewhere.

Referee: 2

Comments to the Author(s)

I found the paper interesting, well-written, and a unique way to use a long-term data set. I was a bit surprised to see that there was significant genetic differentiation over such a (relatively) small area but you did a good job of explaining why the behavioral differences of bears in the sub-population could lead to the observed patterns. I do have some concerns with the statistical tests though as it does not appear that the methods used are appropriate given the range of values possible for most of the dependent variables analyzed (see below for more details). As with so much in statistics, it might be that results derived from more appropriate statistical tests do not differ significantly from the approach originally taken. However, I think it is still important to analyze the data with the correct models to make sure that results are just an artifact of inappropriate models.

One of my biggest comments is to put into better perspective the biological implications of this level of loss of genetic diversity. For example, in your abstract you state that you found "slight but significant genetic differentiation" but throughout the manuscript I didn't really see much discussion about what this level of genetic differentiation really means biologically. Is it really significant enough to lead to negative population-level consequences greater than the effects from causative agent, i.e., sea ice loss? I think this last point deserves a fair amount of discussion to better present the results of this study. I just wonder if loss of genetic diversity is really overly concerning given that polar bears generally don't exhibit that much genetic diversity, globally, to begin with compared to other species and ursids. Given that the same factor that leads to loss in gene flow also has been shown to lead to reduced survival and productivity, won't those have a greater and sooner impact on populations than reduced gene flow and genetic diversity? It just seems like the time scale for genetic effects to manifest themselves into negative population-level consequences is much longer than the direct effects of sea ice loss leading to longer periods fasting on land, reduced access to prey, etc. Again, I'm not discounting the importance of considering the consequences of sea ice loss on polar bear genetics, but would just like it to be put into better context with other factors that are likely to be stronger drivers of population dynamics related to sea ice loss.

Specific comments:

L75: Delete "is not"

L79-80: I am not disputing your results, but I think it is worth further discussion the fact that both the pelagic and "local" bears continue to overlap during the mating season thus calling into question the mechanism presented in the rest of the paper for why there is genetic differentiation occurring. It shouldn't matter if the two groups are isolated during other periods of the year outside of the breeding season. Anyway, some additional discussion about why differentiation is occurring even when there is overlap during the breeding season would be valuable.

L189-190: I think it would be more appropriate to treat period as a categorical variables. That comports more directly with the structure of the data. Is there a reason for treating it as continuous? If you continue to use it as a continuous variable, it would be good to see that it doesn't lead to different results if treating it as categorical variable.

L192-193: Please provide results of the correlation between the period category and sea ice extent.

L194: I don't think your use of a simple linear model is appropriate for the analysis of these dependent variables. Each of these dependent variables does not have the proper support to apply a simple linear model without at least some sort of transformation. A simple linear model would require the dependent variables to be able to take values >0 and <0 and none do.

Additionally, some of the variables can only take values that occur between 0-1 necessitating the use of a Beta regression or something else that supports variables that can only range between 0 and 1.

L199-200: Similar to the above comment, I don't think a linear mixed model is the appropriate test for this dependent variable given that its support ranges from 0-1. This needs a Beta-type regression too (or something similar) with a random intercept. And the square-root transformation described below (L207) does not help with the fact that a linear model will allow negative values in the dependent variable even though they are not possible.

L237: While the Poisson distribution is relevant for simulating discrete values as you'd expect for litter sizes, I don't believe it is being appropriately applied here. Drawing from a Poisson distribution with lambda equal to 2 can lead to litter sizes being drawn that are impossible. It actually leads to ~14% of litters being simulated with litter sizes >3 and ~14% with 0 cubs. A better approach would be a multinomial distribution where probabilities for each of 3 litter sizes (1,2,3) would be applied. This forward-in-time simulation also seems a little simplistic. For example, it appears that bears are allowed to mate and reproduce annually, even though there is at least a 2-3 year time-lag between when females re-breed after having a litter. It doesn't appear that this is accommodated currently.

L241-251: I'd like to see some additional text describing what led to these choices in parameter values for the model.

L255-259: Setting aside the issues I have with the choice of model, while AICc scores helped find the model (in your set) with the most support, it does not say anything about how predictive the model is. Did you perform any sort of analysis (e.g., cross validation) to assess the predictability of the model and not just the fit compared to other models in the set tested?

Referee: 3

Comments to the Author(s)

This study analyzed the spatiotemporal genetic variation of polar bears in the Svalbard archipelago and its association with sea ice decline over a period of 22 years. I found the data properly analyzed, the paper well written, and the conclusions justified. This paper provides new and significant insights into the impact of climate change on polar bears and potentially other sea ice dependent species.

The only main comment I have concerns the discussion of the population genetic differentiation. I find the authors possibly overstate (e.g., p. 20, line 383-386), or incompletely discuss (e.g., p. 20, line 401-403), the genetic structure among Svalbard polar bears.

First of all, the genotyped polar bear individuals were divided into four geographical areas. The appendix refers to previous studies of movement ecology and sea ice presence as the reason for this geographic division, but because it makes an important basis for the analyses and because STRUCTURE analysis seems to favor 3 clusters (and not 4) and it shows significant gene flow among at least some areas for some or all the time periods, the background for these four areas could be explained more and better justified in the main text.

The STRUCTURE analysis also seems to recognize significant variability within the prior designated geographic clusters (Fig. 4 and also seen in Fig. S6), suggesting considerable gene flow and little genetic structure across most of the archipelago. I am not convinced this study has sufficient resolution to see that "The four assigned sampling areas were significantly genetically differentiated, indicating restricted gene flow amongst areas" (p. 20, line 385-386), e.g., there appear to be significant gene flow at least among SWS and SES, and again, just 3 clusters were supported by STRUCTURE. This could be discussed in more detail, or the discussion could be more balanced.

However, the increasing genetic differentiation over time, especially with increasing isolation of NWS from the rest, seems clear and is an important finding, and if possible, STRUCTURE plots

for each of the time periods (similar to Fig. S6) could be included in the main text, or perhaps it could be made clearer from their Fig. 4 (e.g., see below comment about sorting of the individual assignment of clusters).

Some additional minor comments:

P. 4, line 74-75: This sentence needs to be corrected: "...when there is no continuous sea ice cover is not surrounding Svalbard..."

P. 7, line 139: It could be mentioned how potential overlap between individuals among spatiotemporal groups are checked (e.g., have mark-recapture been employed to exclude recaptured individuals?).

P. 8, line 161: For clarity, it could also be mentioned here that the dataset comprised 622 individuals (and not 626 as listed here).

P. 20, line 403: Although perhaps briefly mentioned later in the discussion (line 427-428), it would be worth including a little more detail here about how the distinction in climate regimes between northern and southern ranges can drive the genetic differentiation/reduced gene flow, as this is an important finding of this study.

P. 21, line 409-411: Parts of this sentence is missing or need rephrasing.

Figure 2: Please describe what the dots in the plots represent (medians/averages/other of the different summary stats?).

Figure 4: If sorting the fraction of ancestral population (assignment of individuals to clusters) within areas in the plots (e.g., low to high proportion of orange cluster in each time period of NWS), it may be easier to follow the variation in genetic structure within the designated areas and over time.

It looks like the microsatellite genotyping data is made available through the DRYAD Repository and I encourage the authors to make sure the data is in a format that can easily be interpreted and utilized by the community.

Author's Response to Decision Letter for (RSPB-2021-0011.R0)

See Appendix A.

RSPB-2021-1741.R0

Review form: Reviewer 3

Recommendation

Accept as is

Scientific importance: Is the manuscript an original and important contribution to its field?

Excellent

General interest: Is the paper of sufficient general interest?

Excellent

Quality of the paper: Is the overall quality of the paper suitable?

Excellent

Is the length of the paper justified?

Yes

Should the paper be seen by a specialist statistical reviewer?

No

Do you have any concerns about statistical analyses in this paper? If so, please specify them explicitly in your report.

No

It is a condition of publication that authors make their supporting data, code and materials available - either as supplementary material or hosted in an external repository. Please rate, if applicable, the supporting data on the following criteria.

Is it accessible?

Yes

Is it clear?

N/A

Is it adequate?

N/A

Do you have any ethical concerns with this paper?

No

Comments to the Author

All of the points that I raised have been addressed and I think the authors have done a careful and thorough job responding to the reviews and made appropriate revisions. I have no further comments.

Decision letter (RSPB-2021-1741.R0)

12-Aug-2021

Dear Dr Maduna

I am pleased to inform you that your manuscript RSPB-2021-1741 entitled "Sea ice reduction drives genetic differentiation among Barents Sea polar bears" has been accepted for publication in Proceedings B.

The referee(s) have recommended publication, but also suggest some minor revisions to your manuscript. Therefore, I invite you to respond to the referee(s)' comments and revise your manuscript. Because the schedule for publication is very tight, it is a condition of publication that you submit the revised version of your manuscript within 7 days. If you do not think you will be able to meet this date please let us know.

[http://datadryad.org/submit?journalID=RSPB&manu=\(Document not available\)](http://datadryad.org/submit?journalID=RSPB&manu=(Document%20not%20available)) which will take you to your unique entry in the Dryad repository. If you have already submitted your data to dryad you can make any necessary revisions to your dataset by following the above link. Please see <https://royalsociety.org/journals/ethics-policies/data-sharing-mining/> for more details.

Sincerely,
Dr Daniel Costa
mailto: proceedingsb@royalsociety.org

Associate Editor

Comments to Author:

Thank you for the thorough and clear revisions made. Can you please ensure that a description of the dataset is included in the Dryad link?

Reviewer(s)' Comments to Author:

Referee: 3

Comments to the Author(s).

All of the points that I raised have been addressed and I think the authors have done a careful and thorough job responding to the reviews and made appropriate revisions. I have no further comments.

Author's Response to Decision Letter for (RSPB-2021-1741.R0)

See Appendix B.

Decision letter (RSPB-2021-1741.R1)

12-Aug-2021

Dear Dr Maduna

I am pleased to inform you that your manuscript entitled "Sea ice reduction drives genetic differentiation among Barents Sea polar bears" has been accepted for publication in Proceedings B.

Data Accessibility section

Open Access

Paper charges

Sincerely,

Appendix A

NIBIO

NORWEGIAN INSTITUTE OF
BIOECONOMY RESEARCH

To the Associate Editor

Proceedings B

Date: 03.08.2021

NIBIO
PO Box 115,
NO-1431 Ås, Norway
Tel: +47 406 04 100
post@nibio.no
nibio.no

Ent. nr: 988 983 837

Re: Resubmission of manuscript “RSPB-2021-0011”, entitled: “Sea ice reduction drives genetic differentiation among Barents Sea polar bears”.

Response to Editorial comments:

All three reviewers give positive assessments overall. However two raise a number of detailed concerns about the appropriateness of statistical analyses used and model selection (with the latter, a point is raised about what could be simplistic reliance on the lowest AIC value when selecting a model, something that I have wondered about more generally - what exactly to small differences in AIC mean when comparing different models?). As one reviewer notes, carrying out more appropriate statistical analyses may not end up changing the conclusions, but it is still important for it to be done. One reviewer highlights that parts of the text would benefit from being revised to make it more accessible to the more general reader, which I agree is important.

We are pleased that the original submission of our manuscript has been granted an opportunity for revision and the potential for publication in *Proceedings B*. We appreciate the thorough and useful reviews that have helped us to improve our manuscript, and we thank the reviewers for their constructive criticism. Here we present a revised version of our manuscript that hopefully addresses all the reviewers’ concerns. We have conducted the suggested appropriate analyses outlined by the reviewers. Please refer to the respective dialogs below for our responses surrounding linear modelling and forward-in-time analyses. The editor and reviewers will note that we also restructured and edited the text to improve on the flow of logic and the readability for non-specialists. Below we provide a point-by-point response to the reviewers’ comments. **Reviewer comments are given in italics while our answers are given in standard font. Line number referred to here are those of the clean copy of the manuscript.**

Response to Reviewer #1 comments:

1. Lines 133-139. The title of Table S1 includes the number of individuals in each of three age classes. The inclusion of bears ≤ 2 years old makes me wonder whether data from family groups were

NIBIO

included and, if so, how the dependence (shared genetics) between individuals could be expected to influence the results. Similarly, were data from individuals observed in more than one year included and, if so, how might that influence the results? It would be helpful to describe the data more fully, including such aspects as mentioned above, in the Methods.

We have deliberately included all available bears in our study and therefore (some) family groups are present in the data, including some cubs captured together with their mother. We chose this approach to have as much data as possible available and also because recent studies suggest that there is no simple answer to whether or not close relatives should be removed before analysis. Also, we chose this approach because a major prediction is that increasing sea-ice loss might eventually lead to inbreeding and elevated numbers of close relatives. We found that the mean population-level relatedness increased through time, concordant with our prediction. We confirm that all the bears used in our study are unique, and duplicates had been removed from the dataset prior to the analyses. Given that there was more than 6 years between first and last capture for many individuals, bears captured several times were assigned to the spatiotemporal sample in which they were captured last. Due to the strict word limit in the main text, we have provided a full description of the data in the Appendix S1 (please see **Appendix S1 Lines 34–38**).

2. Lines 164-184. The text in these two sections is probably only fully understandable to a population geneticist. Please consider your non-geneticist readers and add some detail about what these statistics measure and how their use contributes to the objectives of the study. Also, it is probably important to understand whether the various statistics are computed on the basis of an individual bear, a pair of bears, or a sample of bears. If other than a sample of bears, how is the information generalized to a sampling area and period?

We have updated these two sections in the revised manuscript to make it clearer why these statistics are relevant for the objectives of the study. Due to space limitations within the main text, this has been partly solved by adding detailed descriptions of what these various statistical parameters actually measure in Appendix S1. We have also made it clear that all of these descriptors are calculated per sampling site and period, thus facilitating the further statistical analyses of potential spatial and temporal genetic changes due to sea-ice loss which are focal to the study (please see **Appendix S1 Lines 114–161**).

3. Lines 180-181. What are the permutations based on; permutations of what?

Permutations were based on random resampling of individuals. We have reformulated the sentence to make this clearer and moved the revised text to the Appendix S1 (please see **Appendix S1 Lines 158–159**).

4. Lines 194-196. It is unwise to simply select the model with lowest AIC. For example, models with slightly larger AIC having fewer parameters would generally be preferred.

We highly appreciate this comment as it made us look closer into the alternative models, which provided additional insight. Initially, we calculated the *AICc* for all possible models, as it provides an established and objective approach to identify the simplest models that still capture the main patterns

NIBIO

in the data, which was almost invariably a model including only time as a continuous variable. However, a closer inspection of alternative models, containing also study area as predictor variable, suggested not only temporal changes, but also significant geographic differences. Specifically, they showed significantly lower genetic diversity and higher relatedness among polar bears from NWS (north-western Svalbard), where sea ice loss has been particularly severe, compared to bears from the rest of the Svalbard Archipelago. The increased complexity of these alternative models was severely penalized by AIC_C when modelling our dataset of only 16 data points, representing the summary statistics for each combination of study area and time period. In the revised manuscript, therefore, we have also included the results from several alternative models and added corresponding plots of the results for visualization (please see **Manuscript Lines 166–169; Appendix S1 Lines 184–196**).

5. Lines 197-198. Why are differences between the AICs of competing models important? The text in the preceding paragraph states that the model with the smallest AIC was selected as the preferred model, with no mention of model comparison. Does this sentence apply to the mixed models that seem to be the main subject of this paragraph? If so, consider moving this sentence to later in the paragraph and adding an explanation of how model comparisons contributed to the analysis.

ΔAIC , difference in AIC to the best model, allows a quick comparison and ranking of candidate models and is also used for computing ω_i to measure the strength of evidence in favour of a given model. The text belongs to the preceding paragraph where we mention model comparison. We have revised the text accordingly, as this applies to the mixed models as well (please see **Manuscript Lines 166–169; Appendix S1 Lines 184–196**).

6. Lines 201-207. Please clarify this section of text. For example, if measuring differentiation between the same four areas, why would levels of differentiation between areas change? Is the differentiation between two areas not fixed? Also, how are the reduced data sets being formed? What is the larger collection of data from which subsets are being selected?

Please note that pairwise genetic differentiation between areas/sampling populations is never fixed but changes across time and space due to random genetic drift, and that the level of differentiation is negatively correlated with connectivity or genetic exchange. Thus, as gene flow between areas/populations goes down, they will become increasingly differentiated from each other due to a relatively higher influence of genetic drift. This concept is underlying for all genetic monitoring of natural populations, for instance when studying genetic isolation and differentiation due to habitat fragmentation. As we revised the regression analysis and now use a mixed model approach, we removed the analysis involving resampling (please see **Manuscript Lines 154–169**).

7. Lines 212-218. Please expand this section so that a non-geneticist can understand it. What is the LOCPRIOR model? The three-level hierarchy is not clear. Why is it necessary to account for kin structure? What is K ?

In the revised manuscript, we have reformulated this section to make it more understandable: We have included an explanation of what K is, and have explained in more plain words the three-level hierarchy and why it is necessary to account for kin structure. The latter was done to check if kin structure

potentially affected the results, as the data likely included some family groups. However, this made no difference for the results (please see **Manuscript Lines 178–181**).

8. Line 220-231. The first sentence of this paragraph is an example of how the Methods could be improved throughout; most readers will understand a genetic bottleneck and that relatively non-technical information is used to motivate the tests, which most readers will otherwise not understand. Please add more of this sort of information to this section and the preceding sections to improve the understanding of general readers. For clarity, is the range contraction of subpopulations and reduced gene flow between subpopulations what could result in a bottleneck? Is subpopulation synonymous with sampling area?

In the revised manuscript, we have updated the Methods throughout to facilitate a better experience and understanding for specialist as well as non-specialist readers, by cutting down on technical information and by explaining the underlying motivation for the various analyses. Yes, the combinatorial effects of range contraction accompanied by reduction of population size and/or fragmentation of subpopulations and reduced gene flow could result in a bottleneck. And no, a subpopulation is not necessarily synonymous with a sampling area. We understand the confusion here given that the Polar Bear Specialist Group divided the global polar bear population into semi-discrete subpopulations based on telemetry data, geographic barriers, genetics, and site fidelity. By contrast, a genetically defined subpopulation is based on genetic differentiation and genetic clustering analysis. From a population-genetic standpoint, a subpopulation is a portion of the total (global) population that experiences restricted gene flow from other parts of the total population such that its allele frequencies evolve independently to some extent via natural selection (in heterogeneous landscapes *i.e.*, local adaptation) or genetic drift.

9. Line 233. What is NA? I don't think that quantity has been introduced.

This is a typo. We have corrected this to read A_N (please see **Manuscript Line 184**).

10. Line 236. Was sex stochastically determined?

No. We had used the default sex ratio of 1:1.

11. Line 237. A Poisson distribution with mean 2 generates an outcome equal to 0 or exceeding 3 with probabilities of approximately 0.14 (each). That is not a suitable model for polar bears. Most subpopulations of polar bears have either 1 or 2 offspring, with triplets being rare. It would seem that having too many large litters would substantially hasten the loss of genetic diversity; is that correct? How useful are these simulation results when the reproductive model is so unrealistic?

We thank the reviewer for noting that the respective parameters of the model we applied may not be optimal. We have rerun our analysis using NEMO-AGE, the age-/stage-structured version of NEMO, released while our manuscript was undergoing peer-review. NEMO-AGE allows for coding a simplified but realistic demographic model of polar bears amounting to the complete model (detailed in Appendix S1).

NIBIO

12. Line 240. How do the reproductive model (Poisson mean 2) and the population size model (constant or exponential decline) coexist?

In NEMO, the reproductive model we used had Poisson distributed contributions from each parent and all individuals were semelparous. Moreover, NEMO sets an absolute carrying capacity where the per-generation carrying capacity is usually slightly (c. 1–10%) below this based on previous simulation results.

13. Line 241. Is my interpretation that $m = 0.5$ means that individual bears have a 50% probability of moving to a different patch each year correct? If so, how was the population size held constant with what appears to be stochastic migration between patches (similar to the preceding question)? Why was this value chosen; is it a realistic value for the population at this point in time?

In the case of forward-in-time simulations based on the life cycle we used, m represents the forward migration rates, that is the probability of moving to a different patch. So, yes, $m = 0.5$ means that individual bears have a 50% probability of moving to a different patch, but, at each generation in our case. We chose this value to establish a baseline for the different scenarios that we evaluated, and it is not necessarily a reflection of the true population at this point in time. We understand the confusion. We have rephrased ‘model’ to ‘scenario’ given that it is not possible to conduct hypothesis testing from these analyses. All values were chosen based on what an ideal system would constitute and on what we anticipate these estimates will likely be in future. We outline the theoretical genetic framework that facilitated the choice of the different values in Appendix S1 under the section ‘Forward-in-time simulations’ (please see **Appendix S1 Lines 278–295**).

14. Line 243. Is there any rationale for this pattern in the decline in migration rates, such as somehow linking m to projections of sea ice loss?

Yes, these values were chosen based on future prediction on sea-ice extent in the Barents Region and consequent large areas of open water as a barrier to gene flow.

15. Lines 243-244. Please clarify whether g_i refers to generation i . A intermediate model in which a low level of migration persisted might have been interesting. Is there any realistic expectation that gene flow might not reach 0 even if sea ice disappears? For example, do (might) bears swim between areas?

This has been revised accordingly (please see **Appendix S1 Line 303**). Given the spatial arrangement of the Svalbard Archipelago, although physiologically costly, it is possible that bears can swim to other areas and contribute their genes to the gene pool. We now modelled an intermediate model, setting a low migration rate of 1% to account for the possibility that bears can potentially swim to nearby patches.

16. Line 246. Is the 0.975 the lamda parameter in an exponential decay model? Is this an annual or generational rate? What is the expected number of bears left after 120 years? Is the decay stochastic or deterministic?

NIBIO

Yes, it is an exponential decay model reflecting generational rates, and in the case of NEMO the decay is deterministic (see dialog 12). After 120 years, based on the initial carrying capacity of 100, there would be three bears left at each patch.

17. Lines 249-252. Please clarify the fourth model. In particular, how was the empirical genetic data used to parameterize the model? Also, are the rates of gene flow constant from g0 to g5?

Instead of using simulated genotypic data the scenario uses empirically determined genotypes as input for the genetic component of the scenario. In an attempt to simulate a realistic scenario based on our knowledge and given the results, we held gene flow constant from g0 to g4 and g5 to g10. Within NEMO-AGE, when importing a source population file, the plausible stages that could be coded are adults or offspring or both. In addition to using empirical data, we also use simulations to test out the scenario.

18. Lines 263-264. Consider reorganizing the supplementary material so that the table numbers in the two supplements are numbered differently (e.g., both now have a Table S1. Maybe something like S1-1 and S2-1?).

Amended accordingly.

19. Lines 265-266. Was weighted linear regression mentioned in Methods? Is linear regression or weighted linear regression most suitable for this problem? Do they provide fundamentally different information so that both are needed?

Weighted regression models were previously recommended for comparative purposes as the method can account for the fact that each of the response variable measures are estimated with sometimes substantial error. We previously showed that it was not the case in our study. Our sample sizes were typically large and the standard deviation of each of these measures was < 2 , including the residual variance from our models. We choose to remove the sentence altogether to avoid unwarranted confusion.

20. Line 284. I'm not confident that I understand what the statistics in Table S4 measure, but I'm wondering if it is meaningful to compare two different areas in two different time periods. For example, what does a comparison of NWS_T5 and SWS_T1 reveal?

We understand the uncertainty. The comparison between two different areas in two different time periods allows for tracking of short- and long-term spatio-temporal genetic differentiation among sampling areas. For instance, keeping sampling area SWS and temporal T1 (SWS_T1) as a reference point, and comparing it to the different temporal groups at area NWS would tell you the magnitude and rate of change of genetic differentiation from early to recent time periods. And then SWS_T2 vs NWS_Tx and so forth, as cross-validation for the two areas under consideration. As this gets somewhat complicated if temporal groups are not all represented across sampling areas, we kept the strictest sense of a spatio-temporal genetic differentiation analysis in our linear modelling by comparing sampling population comparison per temporal point.

21. Line 332. I think the reference should be to Table S5.

NIBIO

Agreed. Amended accordingly.

22. Line 334. In Figure S7, why does the effective population size increase in the last few generations and the “95% highest posterior density[HPD] estimates” get so much wider?

Because the inference was a population expansion event but with low support as per 95% HPD.

23. Lines 385-386. Does “significantly genetically differentiated” refers to a statistical significance? If so, consider also addressing practical significance. Are the observed differences meaningful in a biological/ecological context? For example, how do the differences between sampling areas compare within the scope of subpopulation differentiation on a global scale?

The clause “significantly genetically differentiated” refers to both statistical and biological significance in this case, hence the proceeding biological interpretation regarding gene flow. Our study assesses genetic diversity and differentiation at a regional scale, and it is the first of its kind across the study area. At a global scale, polar bears have traditionally been shown to exhibit relatively modest subpopulation differentiation, with G_{ST}/F_{ST} -values typically no larger than the ones observed at much smaller geographical scales in our study (e.g., Paetkau *et al.* 1999; Genetic structure of the world’s polar bear populations, <https://doi.org/10.1046/j.1365-294x.1999.00733.x>). In particular, we observed that the level of differentiation among subpopulations of the polar bears in the Svalbard Archipelago increased substantially with time, with G_{ST}/F_{ST} -values being high even in a global context at the end of the study period. Similar changes may have occurred in other polar bear populations elsewhere, however, there are currently no studies available for comparison. We later in the discussion referred to the Foxe Basin and Baffin Bay polar bear subpopulations to embed our results in a broader/global scale.

24. Line 412. Is the text “conservation implications” the conclusion of the sentence or a heading (as it currently appears)?

The sentence was incomplete, but has now been corrected (please see **Manuscript Lines 382–385**).

25. Lines 430-431. Does the idea that distinct management units are emerging depend on the practical significance of the genetic differentiation between sampling areas (see earlier comment about statistical vs. biological significance)?

Yes. We infer that distinct genetic clusters (management units) have evolved in the region. However, if indeed these management units arose as a result of the reduction of sea ice, management should consider finding avenues to restore population connectivity rather than managing these as distinct units.

26. Line 433. I’ll make a comment here with respect to “crucial refugia”, but it could be made almost anywhere in the Discussion. An aspect that seems to be lacking from the Discussion is a consideration of the likelihood that bears will emigrate to more favorable habitat. I suppose that could still be portrayed as a loss of diversity from Svalbard, although those genes would persist elsewhere.

Valid point, however, this would still be depicted as a loss of diversity from Svalbard and the reduction in population size due to emigration is likely to further result in a bottleneck event. Also, a substantial

NIBIO

number of bears stay in Svalbard year-round, and they are presumably less likely to move out. We have updated the section Discussion accordingly.

Response to Reviewer #2 comments:

One of my biggest comments is to put into better perspective the biological implications of this level of loss of genetic diversity. For example, in your abstract you state that you found "slight but significant genetic differentiation" but throughout the manuscript I didn't really see much discussion about what this level of genetic differentiation really means biologically.

We agree that we have been too vague in regard to the biological significance of the observed genetic changes, which are substantial. We have updated the text throughout with this comment in mind. We believe the revised manuscript is clearer on the matter and has improved because of this.

Is it really significant enough to lead to negative population-level consequences greater than the effects from causative agent, i.e., sea ice loss? I think this last point deserves a fair amount of discussion to better present the results of this study.

That is a very good question, as the demographic and genetic effects of sea ice loss are not going to be independent, but are likely to work in concerted action, with genetic effects being more important over longer time spans. Thus, in the short term, demographic effects are going to be more important, as already documented in several studies. However, the observed effects are large and rapid enough to have negative population-level effects over time, and they are likely directly driven by sea-ice loss. The whole text has been updated to make this clearer in the revised manuscript.

I just wonder if loss of genetic diversity is really overly concerning given that polar bears generally don't exhibit that much genetic diversity, globally, to begin with compared to other species and ursids. Given that the same factor that leads to loss in gene flow also has been shown to lead to reduced survival and productivity, won't those have a greater and sooner impact on populations than reduced gene flow and genetic diversity?

We appreciate the reviewers' curiosity. The various negative impacts of sea ice loss on survival, reproduction, gene flow and genetic diversity will probably work in concert and the loss of genetic diversity and gene flow for a species already characterised by low levels of genetic diversity should be alarming and worrisome. A decrease in genetic diversity and likely inbreeding depression can lead to further reduced survival and productivity, resulting in elevated risks of extinction. The implications of reduced genetic diversity and gene flow for species under climatic selective pressure cannot be neglected.

It just seems like the time scale for genetic effects to manifest themselves into negative population-level consequences is much longer than the direct effects of sea ice loss leading to longer periods fasting on land, reduced access to prey, etc. Again, I'm not discounting the importance of considering the consequences of sea ice loss on polar bear genetics, but would just like it to be put into better

NIBIO

context with other factors that are likely to be stronger drivers of population dynamics related to sea ice loss.

We do not discount stronger drivers of population dynamics related to sea ice loss. However, we emphasize that the genetic consequences of sea ice loss and their implications are understudied. Also, the direct link of these drivers to neutral genetic diversity and gene flow is not easily determined, although in the present study we are able for the first time to show empirically that population-genetic changes due to sea-ice loss are happening fast. Simultaneously, our results provide new insight into ongoing demographic changes following loss of sea ice in the polar bear population of the Svalbard Archipelago, by identifying for instance rapid changes in connectivity and population structure, which are difficult to detect early without using genetic tools.

L75: Delete "is not"

Amended accordingly.

L79-80: I am not disputing your results, but I think it is worth further discussion the fact that both the pelagic and "local" bears continue to overlap during the mating season thus calling into question the mechanism presented in the rest of the paper for why there is genetic differentiation occurring. It shouldn't matter if the two groups are isolated during other periods of the year outside of the breeding season. Anyway, some additional discussion about why differentiation is occurring even when there is overlap during the breeding season would be valuable.

We have tried our best to be clearer about this in the revised manuscript. The core of the matter is that the number of pelagic bears that reach Svalbard during the mating season is declining due to sea-ice loss. This reduction in the influx of bears is presumably causing the observed rapid population-genetic changes, by reducing connectivity between the various areas of Svalbard as well as the Archipelago as a whole. Effective dispersal resulting in gene flow will almost always be lower than dispersal *per se*, and sometimes much lower. With gradually fewer migrating bears connecting the various areas of Svalbard through dispersal and gene flow during the mating season due to sea-ice loss, population-genetic changes are gradually emerging. We infer this given the available baseline data in our dataset and interpret our results based on the movement ecology findings of Lone *et al.* (2018) and other similar studies in the respective sections of the Discussion.

L189-190: I think it would be more appropriate to treat period as a categorical variables. That comports more directly with the structure of the data. Is there a reason for treating it as continuous? If you continue to use it as a continuous variable, it would be good to see that it doesn't lead to different results if treating it as categorical variable.

We choose to treat period as continuous because we consider it an ordinal variable and, in doing so, the correlation with sea-ice extent is not lost. However, the point is valid, so in the revised manuscript, we have added plots of the data with time period expressed also as a factor together with a description of the results to show that the main conclusions do not change depending on the approach used (please see **Manuscript Figures 2 & 3**).

NIBIO

L192-193: Please provide results of the correlation between the period category and sea ice extent [SIE].

The correlation based on data presented on Fig.1c was $R = -0.543$, however, when using the mean SIE for each period the correlation was $R = -0.93$. We opted to present the latter within the manuscript, highlighting the importance of temporal effects of sea ice extent.

L194: I don't think your use of a simple linear model is appropriate for the analysis of these dependent variables. Each of these dependent variables does not have the proper support to apply a simple linear model without at least some sort of transformation. A simple linear model would require the dependent variables to be able to take values >0 and <0 and none do. Additionally, some of the variables can only take values that occur between 0-1 necessitating the use of a Beta regression or something else that supports variables that can only range between 0 and 1.

We appreciate this comment and have re-analysed the data using the appropriate methods for each specific variable, including beta regression for genetic estimates that range between 0 and 1. In all cases, we also updated the analyses by using a mixed model framework to control for the fact that our data consisted of repeated estimates across time from the same areas, causing temporal non-independence among data points. In all cases, we obtained identical results. We report on using the beta regression where appropriated in Appendix S1 under the section 'Statistical regression modelling' (please see **Appendix S1 Lines 163–183**).

L199-200: Similar to the above comment, I don't think a linear mixed model is the appropriate test for this dependent variable given that its support ranges from 0-1. This needs a Beta-type regression too (or something similar) with a random intercept. And the square-root transformation described below (L207) does not help with the fact that a linear model will allow negative values in the dependent variable even though they are not possible.

We agree, and we now use a beta mixed regression model. We used unstandardised G''_{ST} as the response variable, period as the fixed effect and the six possible pairs of sampling areas per period as random intercept. A detailed description can also be found in Appendix S1 under the section 'Statistical regression modelling' (please see **Appendix S1 Lines 163–183**).

L237: While the Poisson distribution is relevant for simulating discrete values as you'd expect for litter sizes, I don't believe it is being appropriately applied here. Drawing from a Poisson distribution with lambda equal to 2 can leading to litter sizes being drawn that are impossible. It actually leads to ~14% of litters being simulated with litter sizes >3 and ~14% with 0 cubs. A better approach would be a multinomial distribution where probabilities for each of 3 litter sizes (1,2,3) would be applied. This forward-in-time simulation also seems a little simplistic. For example, it appears that bears are allowed to mate and reproduce annually, even though there is at least a 2-3 year time-lag between when females re-breed after having a litter. It doesn't appear that this is accommodated currently.

Indeed, this was oversight on our part. Unfortunately, NEMO and the newly developed age-/stage-structured version of this software, NEMO-AGE, does not implement a multinomial distributed reproductive model. In our re-analysis, using NEMO-AGE, we used a reproduction model with (i) a

NIBIO

normal distribution (mean of 2 and standard deviation of 0.5) and (ii) a fixed distribution (fecundity of each female is equal to the mean of 2). We found no evidence to suggest that the type of distribution used for the reproductive model had a significant influence on the results, at least in our case. We opted to report the results generated from a reproduction model with a normal distribution (mean of 2 and standard deviation of 0.5). Although NEMO-AGE accommodates age-/stage-structured populations, the realistic demographic model of polar bears is too complex and could not be coded within the program. Instead, we used a simplified yet realistic version of the demographic model of polar bears amounting to the complex model (detailed in Appendix S1) (please see **Appendix S1 Lines 243–295**).

L241-251: I'd like to see some additional text describing what led to these choices in parameter values for the model.

We provide the requested description in Appendix S1 under the section 'Forward-in-time simulations' (please see **Appendix S1 Lines 243–295**).

L255-259: Setting aside the issues I have with the choice of model, while AICc scores helped find the model (in your set) with the most support, it does not say anything about how predictive the model is. Did you perform any sort of analysis (e.g., cross validation) to assess the predictability of the model and not just the fit compared to other models in the set tested?

We appreciate this important input. We did not perform cross validation or other analysis to assess predictability, because our dataset is small and predicting was somewhat out of the scope of this part of the analysis. We rather focused on providing a solid descriptive statistical model of our dataset, and then focus on forward in time predictions using simulations. In the revised manuscript, we have however updated the results section by also presenting perspectives and results from other candidate models than the one preferred by AIC_C . This turned out to be important, because in several cases AIC_C preferred a model that omitted a significant effect of sampling area. We have therefore added plots showing area effects for all variables and statistics for alternative models containing not only sampling period but also sampling areas as predictor variables. We believe this has contributed to improve the manuscript substantially. Additionally, we used different groups to validate our results as outlined in Appendix S1.

Response to Reviewer #3 comments:

The only main comment I have concerns the discussion of the population genetic differentiation. I find the authors possibly overstate (e.g., p. 20, line 383-386), or incompletely discuss (e.g., p. 20, line 401-403), the genetic structure among Svalbard polar bears.

We have revised our manuscript and refined the respective discussion points for clarity and completeness.

First of all, the genotyped polar bear individuals were divided into four geographical areas. The appendix refers to previous studies of movement ecology and sea ice presence as the reason for this geographic division, but because it makes an important basis for the analyses and because

NIBIO

STRUCTURE analysis seems to favor 3 clusters (and not 4) and it shows significant gene flow among at least some areas for some or all the time periods, the background for these four areas could be explained more and better justified in the main text.

Thank you for pointing out that the background for using the four geographical areas as the basis for the genetic STRUCTURE analysis was unclear. Generally, to create the background for which to apply not only STRUCTURE, but all of the various different tests and genetic summary statistics used in this paper to assess spatiotemporal genetic changes, we tried to strike the balance between sample size, number of locations, and number of time points. We ended up with four areas that also had a fair number of temporal points, each with a reasonable sample size that would ensure robust estimates. In the STRUCTURE analysis *per se*, using sampling location/time as prior was useful because it increased the power to detect subtle genetic changes and subtle genetic structure, which we expected might be the case in our polar bear population based on previous studies. However, please note that population-genetic structure is not *a priori*, and the geographical subdivisions should not be confused with the number of genetic clusters. The three genetic clusters are there also when not using information on where the samples were collected, but using this spatial information allows the program to assign individuals with higher probability, providing a more resolved picture. In effect, this allowed us to document with high probability that the three genetic clusters gradually emerged and became unique during the course of the study as the sea ice declined. We have tried our best to be clear about the rationale behind the groupings in the revised version of the manuscript

The STRUCTURE analysis also seems to recognize significant variability within the prior designated geographic clusters (Fig. 4 and also seen in Fig. S6), suggesting considerable gene flow and little genetic structure across most of the archipelago. I am not convinced this study has sufficient resolution to see that “The four assigned sampling areas were significantly genetically differentiated, indicating restricted gene flow amongst areas” (p. 20, line 385-386), e.g., there appear to be significant gene flow at least among SWS and SES, and again, just 3 clusters were supported by STRUCTURE. This could be discussed in more detail, or the discussion could be more balanced.

We provide further discussion and clarity within the main text as suggested by the reviewer. See also our answer to the previous comment. Basically, we show that genetic structure emerges over time and therefore the genetic clusters are clearly delineated only during the last part of the study period. Thus, averaged across all areas and periods (“traditional” STRUCTURE analysis), the genetic clusters are clearly there, but in reality this is a system in rapid change.

However, the increasing genetic differentiation over time, especially with increasing isolation of NWS from the rest, seems clear and is an important finding, and if possible, STRUCTURE plots for each of the time periods (similar to Fig. S6) could be included in the main text, or perhaps it could be made clearer from their Fig. 4 (e.g., see below comment about sorting of the individual assignment of clusters).

Yes, we agree, and the changes observed are not independent of the spatial pattern (see also our answer to the previous two comments). Due to word-count restrictions we could not bring Fig. S6 (now Fig.

NIBIO

S2-5) into the main text, although we modified Fig. S2-5 factoring in the reviewer specifications, by sorting the individuals according to cluster and time period. Additionally, we present further details about G''_{ST} estimates across areas and periods, which complements the STRUCTURE analysis by also documenting changes in genetic structure/differentiation (Fig. 3).

P. 4, line 74-75: This sentence needs to be corrected: "...when there is no continuous sea ice cover is not surrounding Svalbard..."

We have now corrected the sentence.

P. 7, line 139: It could be mentioned how potential overlap between individuals among spatiotemporal groups are checked (e.g., have mark-recapture been employed to exclude recaptured individuals?).

We confirm that all the bears used in our study are unique, and duplicates had been removed from the dataset prior to the analyses. Given that there was more than 6 years between first and last capture for many individuals, bears captured several times were assigned to the spatiotemporal sample in which they were captured last as detailed in Appendix S1.

P. 8, line 161: For clarity, it could also be mentioned here that the dataset comprised 622 individuals (and not 626 as listed here).

We have added the clarifying text.

P. 20, line 403: Although perhaps briefly mentioned later in the discussion (line 427-428), it would be worth including a little more detail here about how the distinction in climate regimes between northern and southern ranges can drive the genetic differentiation/reduced gene flow, as this is an important finding of this study.

Unfortunately, due to word-count restrictions we could not expand the discussion here, but we touch upon the subject later, when we discuss the increasing isolation of NWS from the rest, as the reviewer earlier noted.

P. 21, line 409-411: Parts of this sentence is missing or need rephrasing.

We have added the missing information to complete the sentence.

Figure 2: Please describe what the dots in the plots represent (medians/averages/other of the different summary stats?).

We provide a description that these are mean values in the figure caption.

Figure 4: If sorting the fraction of ancestral population (assignment of individuals to clusters) within areas in the plots (e.g., low to high proportion of orange cluster in each time period of NWS), it may be easier to follow the variation in genetic structure within the designated areas and over time.

Agreed. We have done this for the supported $K=3$ and integrated it into the revised Figs. S2-6.

It looks like the microsatellite genotyping data is made available through the DRYAD Repository and I encourage the authors to make sure the data is in a format that can easily be interpreted and utilized by the community.

NIBIO

The description of the data and its use can already be found within the DRYAD Repository for this manuscript.

--- End of reviews ---

We thank all three anonymous reviewers for investing the time to evaluate our manuscript, and for their positive assessment and constructive feedback. All co-authors have taken part in the revision of the manuscript and are fully responsible for its contents.

Yours sincerely,

Simo N. Maduna and Snorre B. Hagen, on behalf of all the co-authors.

Appendix B

NIBIO

NORWEGIAN INSTITUTE OF
BIOECONOMY RESEARCH

To the Associate Editor

Proceedings B

Date: 12.08.2021

NIBIO
PO Box 115,
NO-1431 Ås, Norway
Tel: +47 406 04 100
post@nibio.no
nibio.no

Ent. nr: 988 983 837

Re: Resubmission of manuscript “RSPB-2021-1741”, entitled: “*Sea ice reduction drives genetic differentiation among Barents Sea polar bears*”.

Response to Editorial comments:

Thank you for the thorough and clear revisions made. Can you please ensure that a description of the dataset is included in the Dryad link?.

We are pleased that the revised submission of our manuscript has been accepted for publication in *Proceedings B* with a minor revision. We have included a description of the dataset in the Dryad link DOI <https://doi.org/10.5061/dryad.zpc866t8x>. We have also updated the section ‘Data Availability’ in the main text accordingly. Below we provide a point-by-point response to the reviewers’ comments. **Reviewer comments are given in *italics* while our answers are given in standard font. Line number referred to here are those of the clean copy of the manuscript.**

Response to Reviewer #3 comments:

All of the points that I raised have been addressed and I think the authors have done a careful and thorough job responding to the reviews and made appropriate revisions. I have no further comments.

We are pleased that our revised manuscript was to the satisfaction of the reviewer.

--- End of reviews ---

We thank the editor and reviewers for investing the time to evaluate our revised manuscript.

Yours sincerely,

Simo N. Maduna and Snorre B. Hagen, on behalf of all the co-authors.